# CDK6 inhibits de novo lipogenesis in white adipose tissues but not in the liver

Alexander J. Hu[1,2], Wei Li[1,3], Calvin Dinh[1], Yongzhao Zhang[1], Jamie K. Hu[1,4], Stefano G. Daniele[5], Xiaoli Hou[1,6], Zixuan Yang [1,7], John M. Asara[8], Guo-fu Hu [1], Stephen R. Farmer [9] & Miaofen G. Hu [1] ✉

Increased de novo lipogenesis (DNL) in white adipose tissue is associated with insulin sensitivity. Under both Normal-Chow-Diet and High-Fat-Diet, mice expressing a kinase inactive Cyclin-dependent kinase 6 (*Cdk6*) allele (*K43M*) display an increase in DNL in visceral white adipose tissues (VAT) as compared to wild type mice (*WT*), accompanied by markedly increased lipogenic transcriptional factor Carbohydrate-responsive element-binding proteins (CHREBP) and lipogenic enzymes in VAT but not in the liver. Treatment of *WT* mice under HFD with a CDK6 inhibitor recapitulates the phenotypes observed in *K43M* mice. Mechanistically, CDK6 phosphorylates AMP-activated protein kinase, leading to phosphorylation and inactivation of acetyl-CoA carboxylase, a key enzyme in DNL. CDK6 also phosphorylates CHREBP thus preventing its entry into the nucleus. Ablation of runt related transcription factor 1 in *K43M* mature adipocytes reverses most of the phenotypes observed in *K43M* mice. These results demonstrate a role of CDK6 in DNL and a strategy to alleviate metabolic syndromes.

Obesity is a risk factor for metabolic diseases, posing a substantial therapeutic challenge towards obesity-related diseases and a large economic burden on the health care[1]. Despite recent advantages in the therapy of obesity and its associated metabolic consequences, obesity-related diseases continue to pose a substantial challenge. It is imperative to develop new effective therapies.

DNL is a process through which fatty acids are synthesized from non-lipid precursors and esterified to form triglycerides (TG). DNL is controlled by a series of metabolic enzymes in the cytoplasm including three rate-limiting enzymes: ATP-citrate lyase (ACLY), acetyl-CoA carboxylase (ACC), and fatty acid synthase (FASN)[2]. Glucose is taken up and converted to citrate through glycolysis and TCA cycle. By

sequential action of ACLY, ACC, and FASN, citrate is converted to acetyl-CoA, malonyl-CoA, and finally palmitate. Stearoyl-CoA desaturase-1 (SCD) then converts palmitate to palmitoleate that mediates the insulin-sensitizing effects of DNL in WAT[3,4]. In addition, DNL in WAT produces other bioactive fatty acids such as fatty acid ester of hydroxyl fatty acids (FAHFAs), a new class of lipids that are important contributors to the improvement of Insulin Resistance (IR)[3–5] and suppression of inflammation in adipose tissues (ATs) by stimulating intestinal cells to secrete glucagon-like peptide 1 (GLP-1), pancreas to release insulin, immune cells to reduce inflammatory cytokine production, as well as enhancing glucose transport in the cells[6–9]. Paradoxically, DNL in WAT is downregulated in obesity[3,5,10] and it is known

[1]Department of Medicine, Division of Hematology and Oncology, Tufts Medical Center, Boston, MA, USA. [2]Department of Surgery, Harvard Medical School, Brigham and Women's Hospital, Boston, MA, USA. [3]Tianjin Medical University Cancer Institute and Hospital, National Clinical Research Center of Cancer, Key Laboratory of Cancer Prevention and Therapy, Tianjin, PR China. [4]University of Miami Miller School of Medicine, Dermatology. 1295 NW 14th St. University of Miami Hospital South Bldg. Suites K-M, Miami, FL, USA. [5]Yale School of Medicine, MD-PhD program, 333 Cedar St, New Haven, CT, USA. [6]Zhejiang Chinese Medical University, Center for Analysis and Testing, 548 Bin-Wen Road, Hangzhou, PR China. [7]TUFTS University Friedman School of Nutrition Science and Policy, TUFTS University, 150 Harrison Avenue, MA Boston, USA. [8]Division of Signal Transduction, Beth Israel Deaconess Medical Center and Department of Medicine, Harvard Medical School, Boston, MA, USA. [9]Boston University School of Medicine, Department of Biochemistry, 72E Concord St, Boston, MA, USA. ✉e-mail: Miaofen.hu@tuftsmedicine.org

that restoring DNL in WAT selectively reverts obesity-induced IR[3,10]. In contrast, DNL in the liver appears to be increased in obesity, and is believed to promote IR, lipotoxicity, and hepatic steatosis[11]. Together, this evidence suggests that reduction of DNL in WAT but its increase in the liver is contributing factors to systemic IR and other metabolic diseases.

Many of the enzymes involved in DNL are regulated primarily at the transcriptional level. Carbohydrate-responsive element-binding proteins (CHREBP), acting as lipogenic transcriptional factors involved in DNL in ATs, are both necessary and sufficient to drive DNL in adipocytes[12]. CHREBP exists as α and β isoforms, with the β isoform being more potent[13] and constitutively active since it lacks most of the N-terminal low glucose-inhibitory domain (LID)[13]. ChREBP is regulated by commuting between the nucleus and cytosol, conformational change, post-transcriptional modification by AMP-activated protein kinase (AMPK) and PKA, and protein degradation[14–16]. In contrast, the sterol regulatory element binding proteins (SREBPs), a family of membrane-bound transcriptional factors identified as important regulators of cholesterol and fatty acid homeostasis[17], are the major players of hepatic DNL[18]. Adipocyte-specific knockout of *CHREBP* dramatically impaired sucrose-induced lipogenic gene expression in both WAT and brown adipose tissue (BAT), which is sufficient to cause IR[6]. Thus, CHREBP is the major determinant of fatty acid synthesis in ATs. However, it remains unknown if there are other regulators that coordinates with CHREBP to activate their target genes[12], since viral re-expression of CHREBP in the liver of SREBP-1c-deficient mice normalized glycolytic gene expression but not lipogenic gene expression[19].

Cyclin-dependent kinase 6 (CDK6), an important cell cycle regulator[20], plays an important function in metabolism[21,22]. To address the role of CDK6 in diseases, we have produced knockout (KO) mice and knock-in mice[23]. The knock-in mutants include a catalytically inactive kinase, CDK6[K43M] (K43M)[23], which imitates pharmacological inhibition of CDK6 kinase activity. Employing these knockout and knock-in mice, we have found that *KO* and *K43M* mice had increased white fat browning, reduced fat mass, improved metabolic profiles, and enhanced insulin sensitivity[22]. They are resistant to HFD-induced obesity. In this study, we found that loss of CDK6 kinase activity enhances DNL in visceral white adipose tissues (VAT) but not in the liver under both NCD and HFD. Inhibition of CDK6 kinase activity recapitulates the phenotypes observed in *K43M* mice. Thus, pharmacological stimulation of DNL in WAT by CDK6 inhibitors is an intriguing possibility for the treatment of diabetes and obesity-related metabolic disorders.

## Results

### Generation of *K43M;Runx1⁻/⁻* (*KR*) mutant mice

Our previous studies have demonstrated that CDK6 has a positive role in induction of obesity by suppressing runt related transcription factor 1 (*Runx1*)[22,24] that is known to be frequently mutated in human leukemia and play a role in hematopoiesis[25]. To genetically ablate *Runx1* in VAT of *WT/K43M* mice (Supplementary Fig. 1), we crossed *WT/K43M* mice with *Runx1* mutant mice *(Runx1[fl/fl])* that bear two *loxP* sites flanking exon 4 of the *Runx1* allele[26]. The resultant *WT;Runx1[fl/fl]* / *K43M;Runx1[fl/fl]* mice were then crossed with *Adipoq-Cre* mice[27] to delete *Runx1* in mature adipocytes[22] (Supplementary Fig. 1). The presence of DNA recombination of *Cdk6* alleles[23,28] (Supplementary Fig. 2a), floxed-*Runx1* alleles[29] (Supplementary Fig. 2b), deleted *Runx1* alleles[29] (Supplementary Fig. 2d) in adipocytes of the VAT from the resultant compound mice but not in the liver and adipose-derived stem cells (ADSC), and *Cre* expression[22] (Supplementary Fig. 2c) were confirmed by PCR. Immunoblotting analysis demonstrated that eWAT still expressed a trace amount of RUNX1 in *KR* compound mice (Supplementary Fig. 2e), probably from the residual RUNX1 protein from adipocyte progenitors and other non-adipocyte cells such as endothelial cells in eWAT. RUNX1 level in the liver and ADSC was not decreased in the compound

*KR* mice, as compared to those in *WT* mice (Supplementary Fig. 2e–g), confirming tissue-specific deletion of *Runx1* in ATs.

### *K43M* mice have increased DNL in VAT but not in the liver

To examine if CDK6 has effects on DNL, we analyzed DNL in conscious mice using stable isotopes coupled to mass spectrometry analysis, which allowed us to assess whole body lipid lipogenesis based on incorporation of radiolabeled products into TG[30]. Following intraperitoneal injection of 10 μCi of D-[¹⁴C(U)]-glucose, blood samples were collected at 0, 5, 10, 30, and 60 min for measurement of D-(U-¹⁴C)-glucose to measure the isotopic enrichment of circulating glucose and the systemic clearance of D-(U-¹⁴C)-glucose (Supplementary Fig. 3). At the end of experiment, e/gWAT and livers were isolated from mice and DNL were examined by measuring D-(U-¹⁴C)-glucose incorporation into organ-specific TG.

In the plasma, even though *K43M* mice on NCD had lower levels of glucose than *WT* mice (Supplementary Fig. 3a, b, e), the average levels of D-(U-¹⁴C)-glucose in the blood of *K43M* and *KR* mice at different time points were either increased significantly compared to male *WT* mice or not significantly different from their female *WT* mice (Fig. 1a, Supplementary 3c, d). We noticed that the incorporation of ¹⁴C from D-[¹⁴C(U)]-glucose into the e/gWAT was elevated by 6.8- and 2.9-fold, respectively, in male and female *K43M* mice than in the corresponding *WT* controls (Fig. 1b, c). In contrast, the incorporation of ¹⁴C from D-[¹⁴C(U)]-glucose into TG of the livers were very similar between *WT* and *K43M* mice (Supplementary Fig. 3f), indicating that loss of CDK6 kinase activity did not increase DNL in the livers. Loss of *Runx1* fully reversed the phenotypes observed in *K43M* tissues (Fig. 1b, c), indicating that RUNX1 is the major downstream effector of CDK6 in DNL.

Next, we examined if upregulation of DNL could contribute to the increased free fatty acids (FFA) and TG in the serum and the liver of *K43M* mice. Besides a marginal decrease in serum TG level (Fig. 1d), no significant change was observed in the level of cholesterol and FFA in the serum or all three in the liver between *K43M* and *WT* mice (Fig. 1e–i). Thus, the increased DNL in VAT did not lead to an increase in serum TG and FFA, indicating an adipose-specific effect of CDK6 in fat metabolism.

### *K43M* mice have increased CHREBP and DNL-associated enzymes

Strong upregulation of DNL in conscious *K43M* mice (Fig. 1) prompted us to study how alteration of CDK6 kinase activity affects the process of DNL at the molecular and cellular level. Since CHREBP is the major determinant of fatty acid synthesis in adipose tissues, we evaluated the expression levels of CHREBP with anti-CHREBP antibody that detects both α (~110KD) and β (~78KD) isoforms (Fig. 2a, b). The protein level of CHREBPα was increased significantly in *K43M* eWAT and liver (Fig. 2a–d), while the protein level of CHREBPβ was increased significantly in *K43M* eWAT but not in the liver, accompanied by a pronounced increase in mRNA (Fig. 2e) and protein (Fig. 2g, i–m) levels of glucose transporter-1 (GLUT1), ACLY, ACC1, FASN, and SCD1 in the eWAT. Although the mRNA levels of *Glut1, Acly, and Scd1* were elevated in *K43M* livers (Fig. 2f), the protein levels were comparable with those in *WT* livers (Fig. 2h, j). Of note, protein level of GLUT1, which is present in all cells to provide basal glucose uptake[31], was markedly increased in *K43M* eWAT (Fig. 2g, i) but not in the liver (Fig. 2h, j), whereas glucose transporter-4 (GLUT4), which is expressed only by cells that accelerate glucose transport in response to insulin[31], was reduced in both eWAT and liver (Supplementary Fig. 4a–d). Similar data were obtained from gWAT (data not shown). Thus, loss of CDK6 kinase activity in e/gWAT resulted in increased CHREBPα and CHREBPβ in VAT and a series of coordinated enzymes involved in DNL under NCD.

Among the known targets of CDK6[22,24,32], RUNX1, an transcriptional factor, has been shown to bind to the promoter region of *SCD1* to modulate fatty acid production in epithelial cells by up-regulating

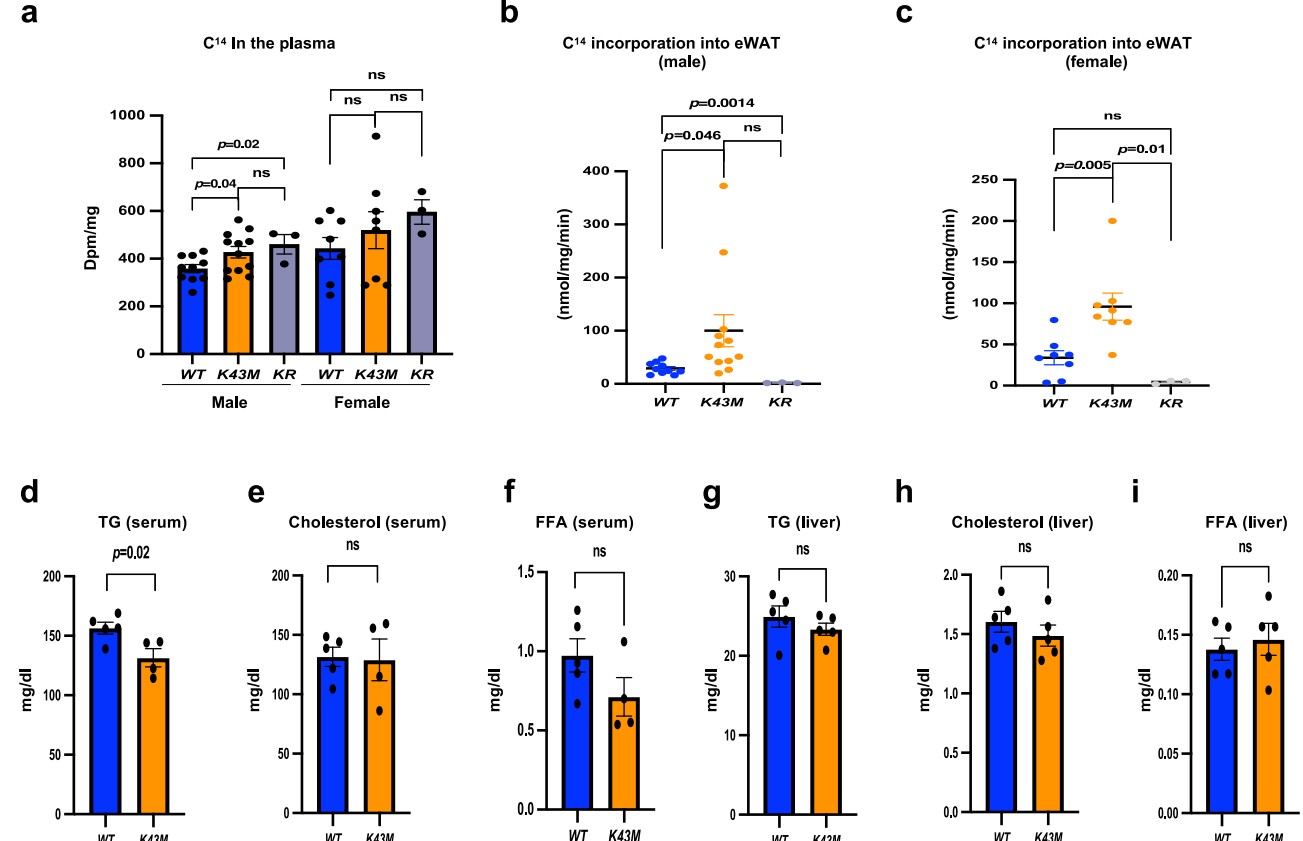

**Fig. 1 | DNL is increased in e/gWAT of *K43M* mice and is reversed upon ablation of RUNX1.** Mice at the age of 6-8 months were i.p. injected with 10 μCi of D-[$^{14}$C(U)]-glucose. The D-(U-$^{14}$C)-glucose counts in the blood **a** were evaluated at 5-, 10-, 30-, and 60-min post-injection of D-[$^{14}$C(U)]-glucose. Bar graphs shown **a** are the average of D-(U-$^{14}$C)-glucose counts from 4 different time points. For male, *WT* (*n* = 10), *K43M* (*n* = 12), and *KR* (*n* = 3). For female *WT/K43M* (*n* = 8), and *KR* (*n* = 3). At the end of experiments, e/gWAT were harvested and extracted to determine D-(U-$^{14}$C) incorporation into TG **b**, **c**. Data were shown **b**, **c** as mean ± SE. For b, *WT* (n = 10),

*K43M* (*n* = 12), and *KR* (*n* = 3); for **c**, *WT/ K43M* (*n* = 8), and *KR* (*n* = 3). **d–f** Serum concentrations of TG **d**, cholesterol **e**, and FFA **f**. **g–i** hepatic concentrations of TG **g**, cholesterol **h**, and FFA **i**. Data were shown as mean ± SE. For **d–f**, *WT* (*n* = 5), *K43M* (*n* = 4); for **g–i**, *WT* and *K43M* (*n* = 5). We calculated statistical significance using two-tailed Student's T-test, with *p* < 0.05 considered significant. The p values were summarized above two compared groups. NS indicates no significance between two groups.

mRNA and protein levels of SCD1 and SCD1-product oleate[33]. In agreement with a previous report[33], we found that loss of *Runx1* in *WT* mice decreased SCD1 level, accompanied by a reduction of the protein levels of ACLY, ACC1, and FASN (Supplementary Fig. 5a, b), ranging from ~4.5- to 10.6-fold reduction in eWAT. Remarkably, loss of *Runx1* fully or partially reversed upregulation of these proteins in *K43M* eWAT (Fig. 2g, i), indicating that RUNX1 is a major downstream effector of CDK6 in negative regulation of DNL.

To examine if RUNX1 can control expression of CHREBP, we measured CHREBP levels in *WT* and *K43M* eWAT in the presence or absence of RUNX1, and found that loss of *Runx1* in *WT* mice did not change the levels of CHREBP (Supplementary Fig. 5c–e), whereas loss of *Runx1* in *K43M* eWAT partially or fully reversed the increase of CHREBP in *K43M* eWAT and liver (Fig. 2a–d), suggesting that RUNX1 does not mediates CHREBP expression in *WT* mice.

### *K43M* mice under HFD have significant increase in lipogenic enzymes in eWAT but not in the liver

To determine if the improved insulin sensitivity in *K43M* mice is correlated with increased DNL, we subjected *WT, K43M* and *KR* mice to a HFD (starting at 4-week-old) over a 14-week period as described[22]. Livers and eWAT were isolated from 18-week-old mice under HFD, and protein expression levels were determined. We found that in HFD-fed animals, CDK6 levels were increased in adipose tissues[22] and liver (Supplementary Fig. 6a) upon HFD feeding. Consistent with previous

studies[34,35], HFD-induce obesity[22] and IR[22] are associated with reduced expression of DNL-related proteins[35] including CHREBP, ACLY, ACC1, and FASN, but increased expression of *SCD1*[34] protein in *WT* mice (Supplementary Fig. 6b). However, the DNL-related proteins were increased dramatically in *K43M* mice compared with those in *WT* mice (Fig. 3a, c) under HFD. A loss of *Runx1* also fully or partially reversed the enhanced expression of ACLY, ACC1, and SCD1 proteins in *K43M* mice in HFD-fed animals (Fig. 3a, c). Of note, the GLUT1 protein levels were not altered significantly among different strains of mice under HFD (Fig. 3). In contrast, these DNL-related proteins were not increased, but decreased in the liver of *K43M* mice, even though only ACLY reached statistically significance (Fig. 3b, d). DNL-related proteins were slightly but not statistically significantly recovered in *KR* livers (Fig. 3b, d), consistent with the fact that *Adipoq-Cre* only delete *Runx1* in mature adipocytes but not in the liver. Thus, the role of RUNX1 in CDK6-mediated DNL in the liver, if any, is unclear currently. Collectively, these results showed that CDK6 is a sensor of nutrients, and loss of CDK6 kinase activity in mice under HFD increased DNL-associated lipogenic enzymes in eWAT but not in the liver. Loss of RUNX1 partially or fully reversed the phenotypes observed in *K43M* eWAT, reinforcing a role of RUNX1 in CDK6-mediated negative regulation of DNL.

### CDK4/6 inhibitor recapitulates the effect of K43M in DNL

To determine if CDK6 inhibitor recapitulates the effect of *K43M* in DNL, we treated *WT* mice under HFD (starting at 4-week-old) with vehicle

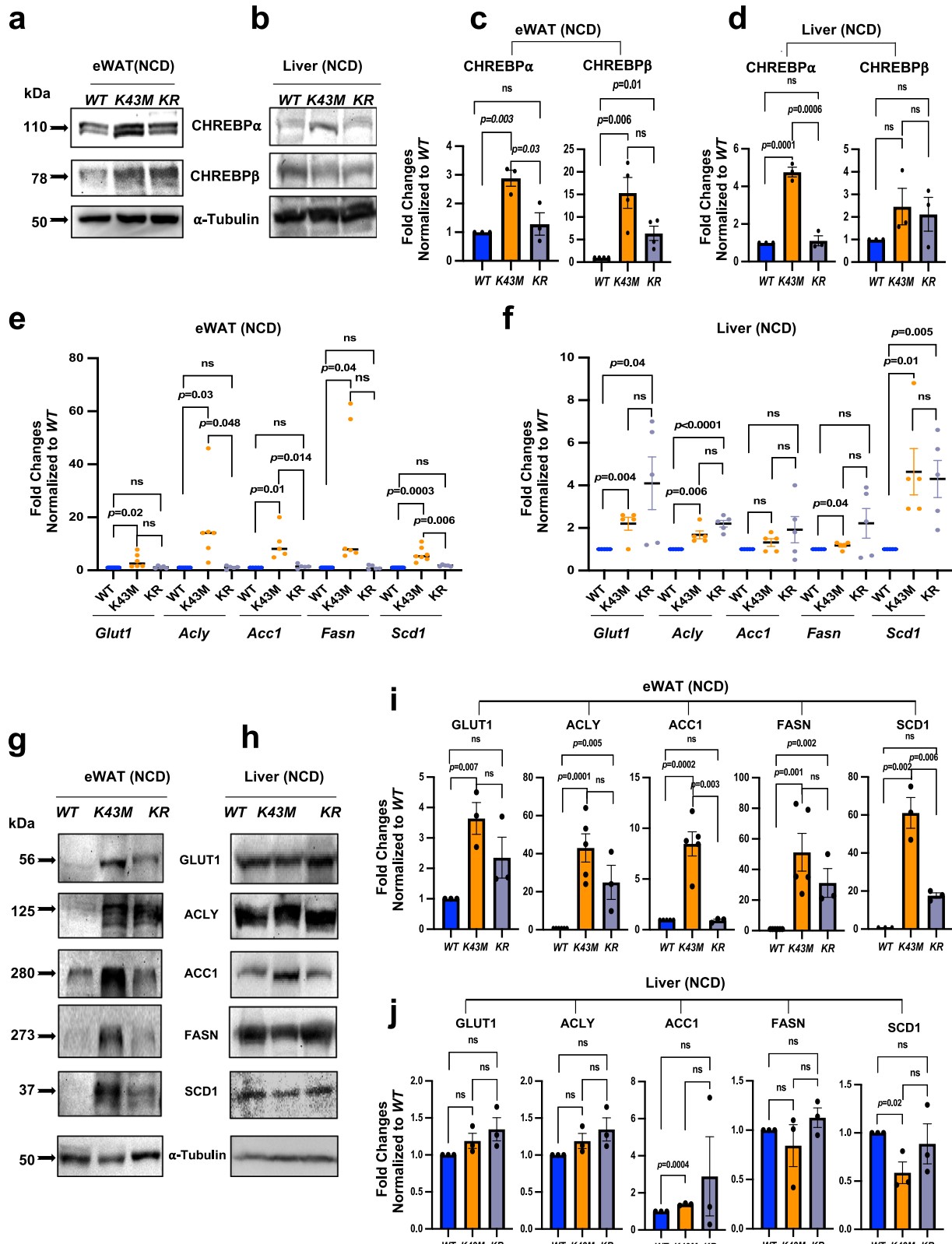

(0.5% methycellulose, V) or LEE011 (Lee) (200 mg/kg daily by gavage)[32], a clinically relevant small molecule inhibitor of CDK4 and CDK6 kinases[36], for 14 weeks beginning at the age of 6 weeks. Both male and female (data not shown) mice displayed a tendency of reduced weight gain under HFD upon treatment of Lee (Fig. 4a). The body weight was reduced by 15%, 17%, and 19%, respectively, after 11-, 12-, and 14-weeks treatment. Consistently, we observed a 2.4-, 2.2-, and 2.5-fold reduction in fat pad mass of iWAT, eWAT, and para-renal WAT, respectively, in treated animals (Fig. 4b, c). No difference in the mass was observed in in the liver and BAT of treated-mice as compared to that of V-treated mice (Fig. 4c). Interestingly, expression of BAT-specific genes *Ucp-1* and *Pgc-1α* was elevated in iWAT upon treatment (Fig. 4d). Furthermore, the protein level of DNL-related enzymes, including ACLY, ACC1, and FASN was increased significantly in eWAT

**Fig. 2 | GLUT1, CHREBP and DNL enzymes are differentially regulated in WAT and liver in *K43M* mice. a, b, g, h** Representative immunoblots of the indicated protein in eWAT or liver from 100 µg of cell lysates of male *WT, K43M, and KR* mice at the age of 18 weeks. α-tubulin was used as loading control. Three-five independent experiments were repeated with similar results. **c, d, i, j** Bar graphs summarizing fold changes of different protein expression from 3 to 5 independent experiments. The intensity of each protein was measured by **FluorChem M** system and then normalized to α-tubulin. The fold change of each protein was normalized to the *WT* control, which was arbitrarily set to 1 unit. Data were shown as mean ± S.E (*n* = 3–5 for different groups). **e, f** Relative mRNA expression levels of *Glut1*, and DNL-specific markers (*Acly, Acc1, Fasn, and Scd1*) of eWAT **e** and liver **f** tissues from male *WT, K43M, and KR* mice (*n* = 5–7 for different groups). The fold changes of each mRNA were normalized to the *WT* control, which was arbitrarily set to 1 unit. Data were shown as mean ± SE. *, We calculated statistical significance using two-tailed Student's T-test, with *p* < 0.05 considered significant. P values were displayed above two groups. NS indicates no significance between two groups.

but not in the liver of treated mice (Fig. 4e, h). Collectively, these results show that inhibition of CDK6 kinase activity in mice under HFD protected from diet-induced obesity, reduced VAT masses, and increased white fat browning in iWAT, accompanied by upregulated DNL in VAT but not in the liver.

One concern here is the specificity of Lee, a dual inhibitor for CDK6 and CDK4[37], which has been shown to be involved in HFD-induced obesity[38]. However, it is unclear if CDK4 has negative role in DNL. To determine if CDK4 has a similar role as CDK6, we performed DNL analysis in vitro by using *WT* and CDK4 deficient mouse embryonic fibroblasts (MEFs)[39], which have been utilized as a surrogate stem cell model for Bone Marrow derived Stem Cell (BMSC) to study differentiation of mesoderm-type cells including adipocytes[40]. Similar as ADSC derived from e/gWAT (Supplementary Fig. 7a), loss of CDK4 in MEFs also increased expression levels of DNL-related genes (Supplementary Fig. 7b) including GLUT1, ACLY, ACC1, FASN, and SCD1. Thus, CDK4 has a similar role as CDK6 in negative regulation of DNL in vitro.

**The role of AMPKα in CDK6-mediated suppression of DNL in ATs**

To delineate the signaling pathways involved in CDK6-mediated suppression of DNL in ATs, we examined the involvement of known downstream regulators of CDK6. CDK6 phosphorylates and inhibits Tuberous Sclerosis Complex 2 (TSC2)[41] to promote mammalian target of rapamycin (mTOR) activity, which drives DNL in BAT[42]. Thus, CDK6 may couple mTOR in regulation of DNL in our mouse models. In agreement with previous studies[43], *K43M* eWAT had reduced levels of phosphorylation of mTOR on S2448 (activation form) and reduced total mTOR protein level, whereas *K43M* liver had increased phosphorylation of mTOR on S2448 and increased mTOR protein level (Fig. 5a–d). Thus, mTOR seems not involved in CDK6-mediated negative regulation of DNL in VAT.

Under the same experimental conditions, *K43M* e/gWAT had reduced levels phosphorylated AMPKα on T172 (p-AMPKα) (Fig. 5e, g), which is required for AMPK activation[44,45], accompanied by the reduced phosphorylation level of ACC1 (S79, p-ACC1) in eWAT but not in the liver (Fig. 5e–h). In contrast, *K43M* liver had slightly but significantly increased levels of p-AMPKα (Fig. 5f, h). Deletion of *Runx1* in *WT* or *K43M* eWAT (Supplementary Fig. 5f-h, Fig. 5e, g) did not significantly change the levels of p-AMPKα, as seen in the levels of mTOR S2448 (Fig. 5a, c), suggesting that loss of RUNX1 is not sufficient to alter CDK6-mediated phosphorylation of AMPKα T172 and mTOR S2448. However, ablation of *Runx1* in *K43M* eWAT fully reversed the reduction of p-ACC1 in *K43M* eWAT (Fig. 5e, g), suggesting that loss of RUNX1 may have led to activation of other kinases which compensated the loss of CDK6 kinase activity on phosphorylation of ACC1, which may explain the hyper-phosphorylation of ACC1, an important rate-limiting enzyme for synthesis of malonyl-CoA, resulting in a reduction in lipid synthesis rates (Figs. 1b, c and 2). Thus, AMPKα may be involved in CDK6-mediated negative regulation of DNL.

**CDK6 interacts with AMPKα and CHREBP and phosphorylates them**

Given the reduced phosphorylation levels of p-AMPKα and increased CHREBP in e/gWAT of *K43M* mice, we investigated if CDK6 interacts and phosphorylates them. We performed co-immunoprecipitation assay to examine if CDK6 interacts with CHREBP and AMPKα. First, we

ensured that the antibodies of CHREBP, CDK6, T*PXK/R, and AMPKα used were able to immunoprecipitate their respective antigens from the cell lysates. The reason to include T*PXK/R was that the consensus CDK phosphorylation motif is present in both CHREBP and AMPKα and that the specific antibodies against p-CHREBP are not available. Normal rabbit IgG was used as the negative control. Figure 6a–d demonstrated that all the tested target proteins were able to bind to their own antibodies but not to IgG, indicating target protein binds to its antibody specifically (Fig. 6a–d).

Next, the function of CDK6 on its targets was examined by protein interaction assays. Cell lysates of eWAT isolated from *WT, K43M*, and *KR* mice were immunoprecipitated with an AMPKα or CDK6 (Fig. 6e, f) antibodies, and the immuno-precipitates were probed with antibodies against CDK6, AMPKα, p-AMPKα, and p-ACC1. As expected, p-ACC1 were co-immunoprecipitated with AMPKα[46,47]. CDK6 was also co-immunoprecipitated with AMPKα. The level of CDK6-bound p-AMPKα and p-ACC1 were drastically reduced in *K43M* eWAT as compared to that in *WT* eWAT. In contrast, CDK6-bound p-AMPKα and p-ACC1 in the eWAT of *KR* mice was comparable to that of *WT* mice, suggesting that CDK6 is not the sole kinase to phosphorylate AMPKα, and that deletion of *Runx1* on the K43M background may lead to activation of other kinases to compensate the loss of CDK6 kinase activity on phosphorylation of ACC1. Indeed, AMPKα can be phosphorylated and activated by serine/threonine kinase LKB1 directly[48] in the absence of CDK6 kinase activity. Of note, in addition to p-AMPKα and p-ACC1, CHREBPβ was also co-immunoprecipitated with CDK6 (Fig. 6f). To confirm the interaction between CDK6 and CHREBPβ, CHREBP was immunoprecipitated (Fig. 6g). We found that CHREBPβ but not CHREBPα was co-immunoprecipitated with CDK6. Together, these results suggest that CDK6 interacts directly with AMPKα and CHREBPβ.

For analysis of direct phosphorylation by CDK6, cell lysates were immunoprecipitated with p-AMPKα and then followed by immuno-blotting with different antibodies as indicated (Fig. 6h). We observed that the levels of p-AMPKα-bound CDK6, CHREBPβ, and p-ACC1 were reduced in the eWAT of K43M as compared to those of *WT*. Consistent with the observations shown in Fig. 6e, f, the levels of p-AMPKα-bound CDK6, CHREBPβ and p-ACC1 in *KR* eWAT were comparable to that in *WT* eWAT. These results suggest that CDK6 phosphorylates AMPKα.

Since there are no commercially available phosphor-CHREBP antibodies, we used an antibody that recognize the phosphor-threonine site in the consensus CDK phosphorylation motif (p-T*PXK/R)[49,50] of the immunoprecipitated phosphorylated proteins from the cell lysates of eWAT and then probed with antibodies against p-AMPKα, CHREBP, and p-ACC1 (Fig. 6i). The phosphorylation profiles of p-AMPKα and p-ACC1 were the same as that observed in Fig. 6e, f, h. Similarly, the levels of p-T*PXK/R bound CHREBPβ in *K43M* eWAT were drastically reduced in *K43M* eWAT as compared to that in *WT* eWAT, while the level of p-T*PXK/R-bound CHREBPβ in *KR* eWAT was comparable to that in *WT* eWAT. To accurately compare the signal between two different antibodies in the same blot, we stripped CHREBPβ antibody off the blot and re-probed with p-T*PXK/R antibody and found the same profiles, confirming that phosphorylation of CHREBPβ was dramatically decreased in *K43M* cell lysates. Together, the data show that CDK6 interact with AMPKα, CHREBP, and may phosphorylate AMPKα and CHREBP directly or indirectly.

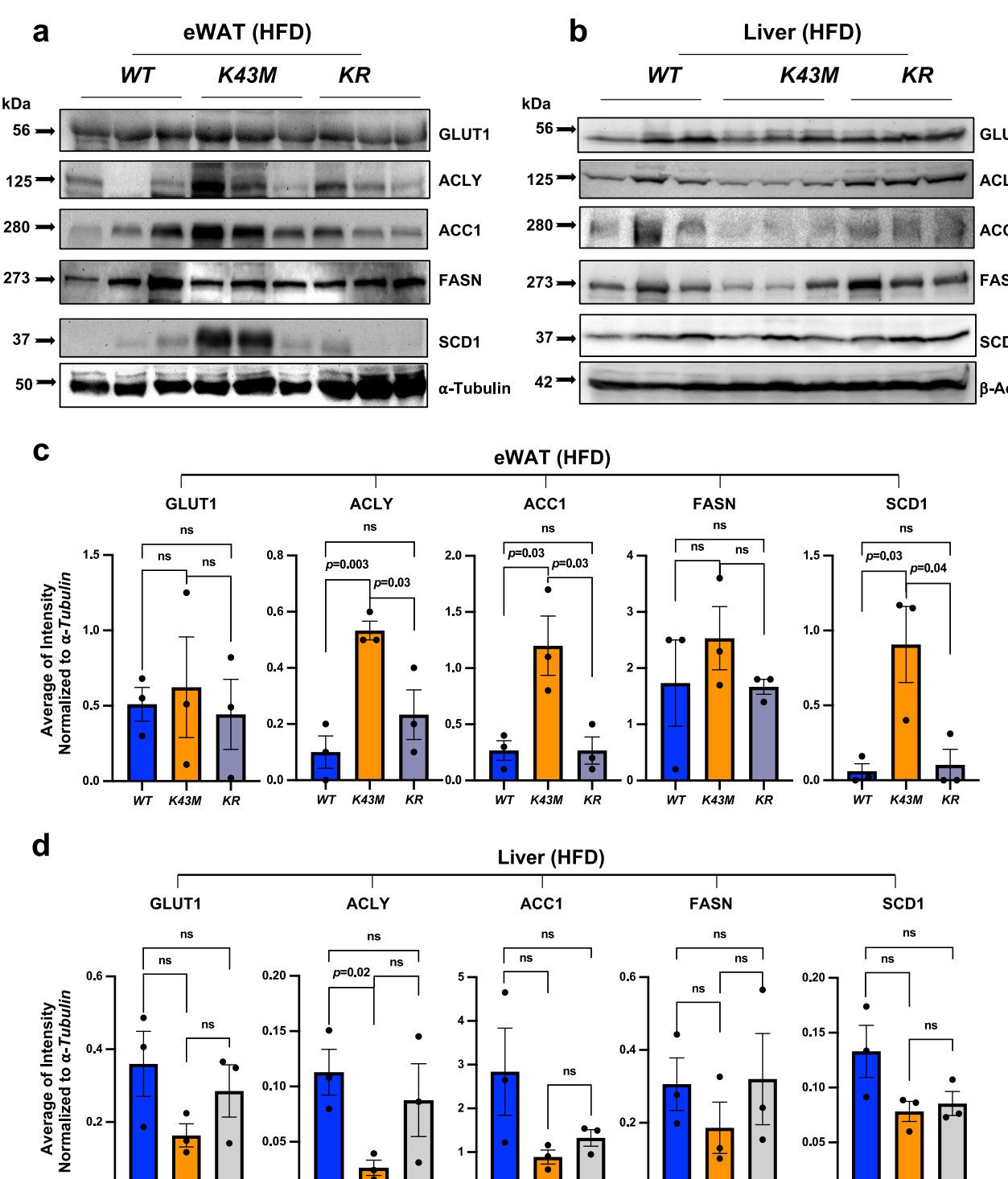

**Fig. 3 | _K43M_ mice under HFD have increased lipogenic enzymes in eWAT but not in the liver. a, b** Representative immunoblots of the indicated protein levels in eWAT **a** or liver **b** from 100 μg of cell lysates of male _WT_ (_n_ = 3), _K43M_ (_n_ = 3), _and KR_ (_n_ = 3) mice under HFD for 14 weeks, starting at the age of 4 weeks (similar data were obtained in female mice, data not shown). α-tubulin/β-actin was used as loading control. Three independent experiments were repeated with similar results. **c, d** Bar graphs summarized fold changes of different protein expression from 3 different mice either from eWAT **c** or from liver **d**. The intensity of each protein was measured by FluorChem M system and then normalized to α-tubulin/β-actin . Data were shown as mean ± SE (_n_ = 3 per group). We calculated statistical significance using Student's two-tailed T-test, with _p_ < 0.05 considered significant. P values were displayed above two groups. NS indicates no significance between two groups.

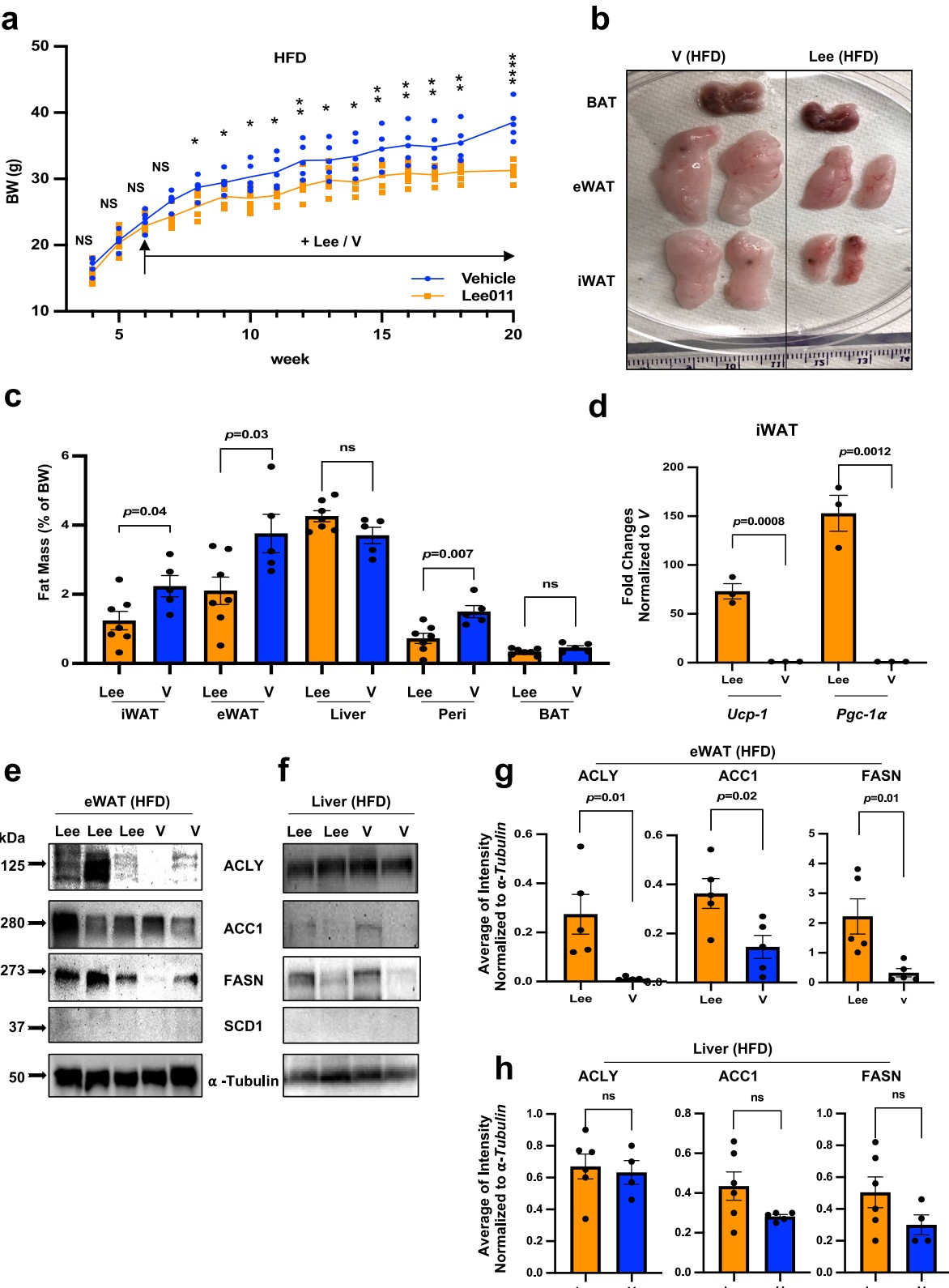

## CDK6/D3 phosphorylates serine and threonine residues of human AMPKα and CHREBP directly

To determine if CDK6 directly phosphorylates CHREBP and AMPKα, we performed in vitro kinase assay followed by proteomic analysis. CHREBP and AMPKα are evolutionarily highly conserved among human and mouse species[14,51]. Human proteins were therefore used for more physiological relevance. Flag-tagged human AMPK-α1, AMPK-α2,

and CHREBP fusion proteins were expressed in 293 T cells and purified with magnetic beads conjugated anti-Flag antibody. The identity of each fusion proteins was confirmed by immunoblotting with anti-Flag antibody (Supplementary Fig. 8a).

The purified immunoprecipitants were divided into two parts, one of which was separated on SDS-PAGE and stained with Coomassie Brilliant R-250 Blue (CBB) (Supplementary Fig. 8b). The stained

**Fig. 4 | CDK4/6 inhibitor prevented HFD-induced obesity, reduced VAT masses, increased white fat browning on SAT, and upregulated DNL in eWAT but not in the liver. a** Body weight of age-matched male mice on HFD for a 16-week observation time. HFD was started at the age of 4 weeks. LEE011 (Lee, $n = 7$) or vehicle (V, $n = 5$) control was administered at the age of 6 weeks for 14 weeks. We calculated statistical significance using two-tailed Student's T-test, with $p < 0.05$ considered significant. *, $p < 0.05$, **, $p < 0.01$, ****, $p < 0.0001$. NS indicates no significance between two groups. All the p values were displayed in source data files under **a**. **b** Appearance of isolated BAT, eWAT and iWAT from V- and Lee-treated mice. **c** Mass of various fat pads was normalized to body weight of male mice on HFD at the age of 20 weeks. For **a**, **c**, data shown were mean ± SE. ($n = 7$ for lee, $n = 5$ for V). **d** Relative mRNA expression levels of BAT-specific markers of iWAT tissues. Data

shown were fold changes of mRNA of Lee-treated ($n = 3$) normalized to those of V-treated control mice ($n = 3$), which was arbitrarily set to 1 unit. **e**, **f** Representative immunoblots of the indicated protein levels in eWAT **e** or liver **f** from 100 μg of cell lysates of male *WT* mice treated with Lee/V under HFD. Five to six independent experiments were repeated with similar results. **g**, **h** Bar graphs summarized fold changes of different protein expression from 4 to 6 independent experiments either from eWAT **g** or from liver **h**. The intensity of each protein was measured by FluorChem M system and then normalized to α-tubulin. Data were shown as mean ± SE ($n = 4$–6 for each group). We calculated statistical significance using two-tailed Student's T-test, with $p < 0.05$ considered significant. P values were displayed above two groups. NS indicates no significance between two groups.

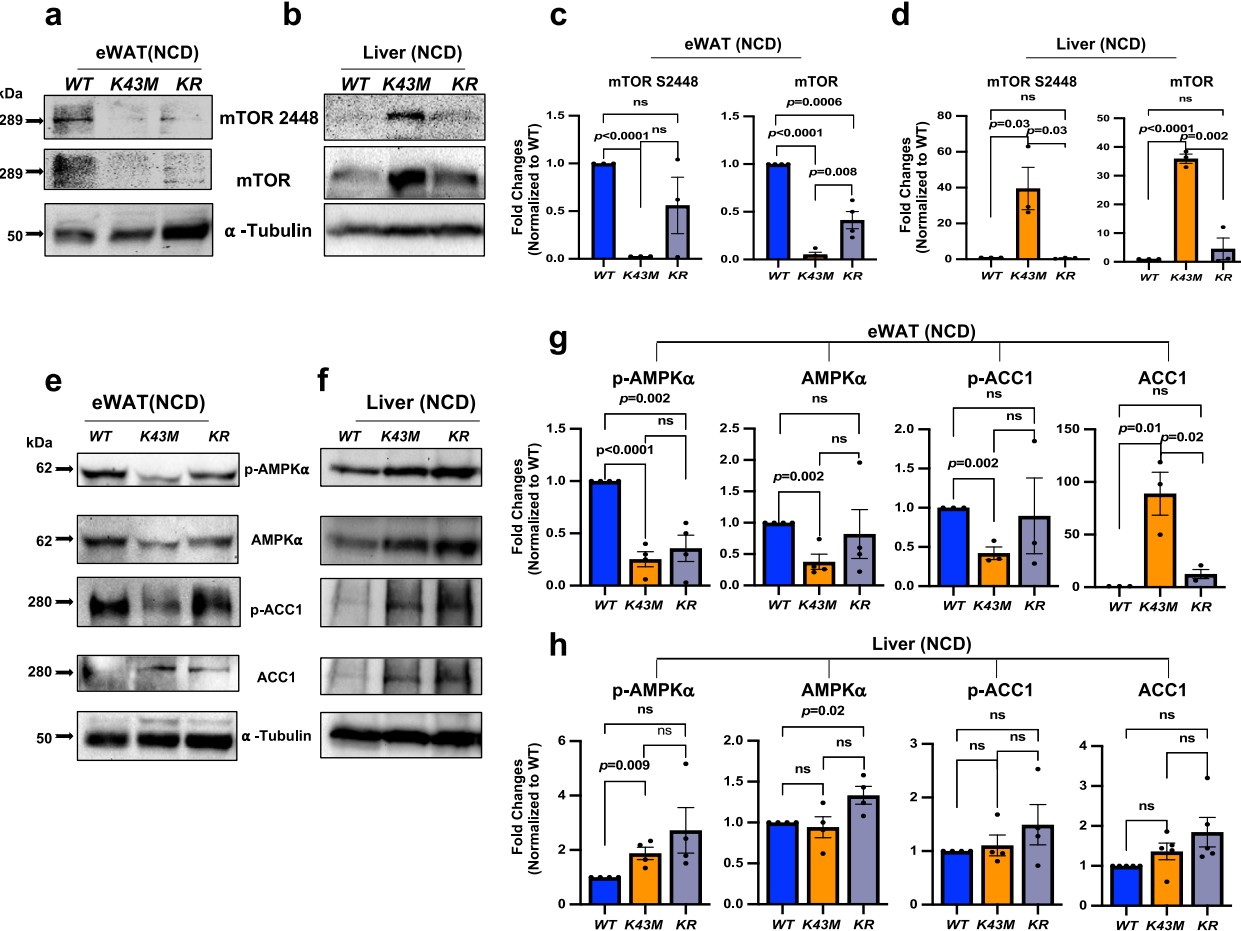

**Fig. 5 | Loss of CDK6 kinase activity leads to reduced phosphorylation of AMPK-α and ACC1 in eWAT, but enhanced phosphorylation and protein levels of mTOR in the liver. a**, **b**, **e**, **f** Representative immunoblots of the indicated proteins in eWAT **a**, **e** or liver **b**, **f** from 100 μg of cell lysates of male *WT, K43M, and KR* mice under NCD for 18 weeks. α-tubulin was used as loading control. **c**, **d**, **g**, **h** Bar graphs summarizing fold changes of different protein expression from 3 to 5 independent

experiments. The intensity of each protein was measured by **FluorChem M** system and then normalized to α-tubulin. The fold change of each protein was normalized to the *WT* control, which was arbitrarily set to 1 unit. Data were shown as mean ± SE ($n = 3$–4 for each group). We calculated statistical significance using two-tailed Student's T-test, with $p < 0.05$ considered significant. *P* values were displayed above two groups. NS indicates no significance between two groups.

proteins (~1 μg) were excised and subjected to phosphorylation analysis by Mass Spectrometry. The other part was used as the substrate for kinase reaction of CDK6/Cyclin D3 (CDK6/D3), and then subjected to SDS-PAGE separation (Supplementary Fig. 8c), followed by Mass Spectrophotometry analysis for phosphorylated peptides.

The total spectrum counts (TSC) of phosphorylated peptides dramatically increased after kinase reaction of CDK6/D3 (Supplementary Fig. 9a). We observed a 6.43-, 14.65-, and 46.75-fold increase in TSC of CHREBP, AMPK-α1, and AMPK-α2, respectively, after reaction with CDK6/D3, indicating that CHREBP, AMPK-α1, and AMPK-α2 are

substrates of CDK6. Multiple CDK6-specific phosphorylation sites were detected in CHREBP (Supplementary Fig. 9b). Among them, S47, S59, S65, and S71 are located within its N-terminus[14,52] (amino acids 1-251) that are known to be important for nuclear export/import and glucose-sensing[53] (Fig. 7a); Others including T335, S361, S369, S449, S516, S520, S614, S619, and S631 were dispersed across different domains (Fig. 7a). In addition, we have also found two phosphorylation sites which are non-CDK6-specific, including S196 (PKA-dependent)[54] and S602, since the phosphorylation was observed both with or without reactions to CDK6/D3 (Supplementary Fig. 9b). Moreover,

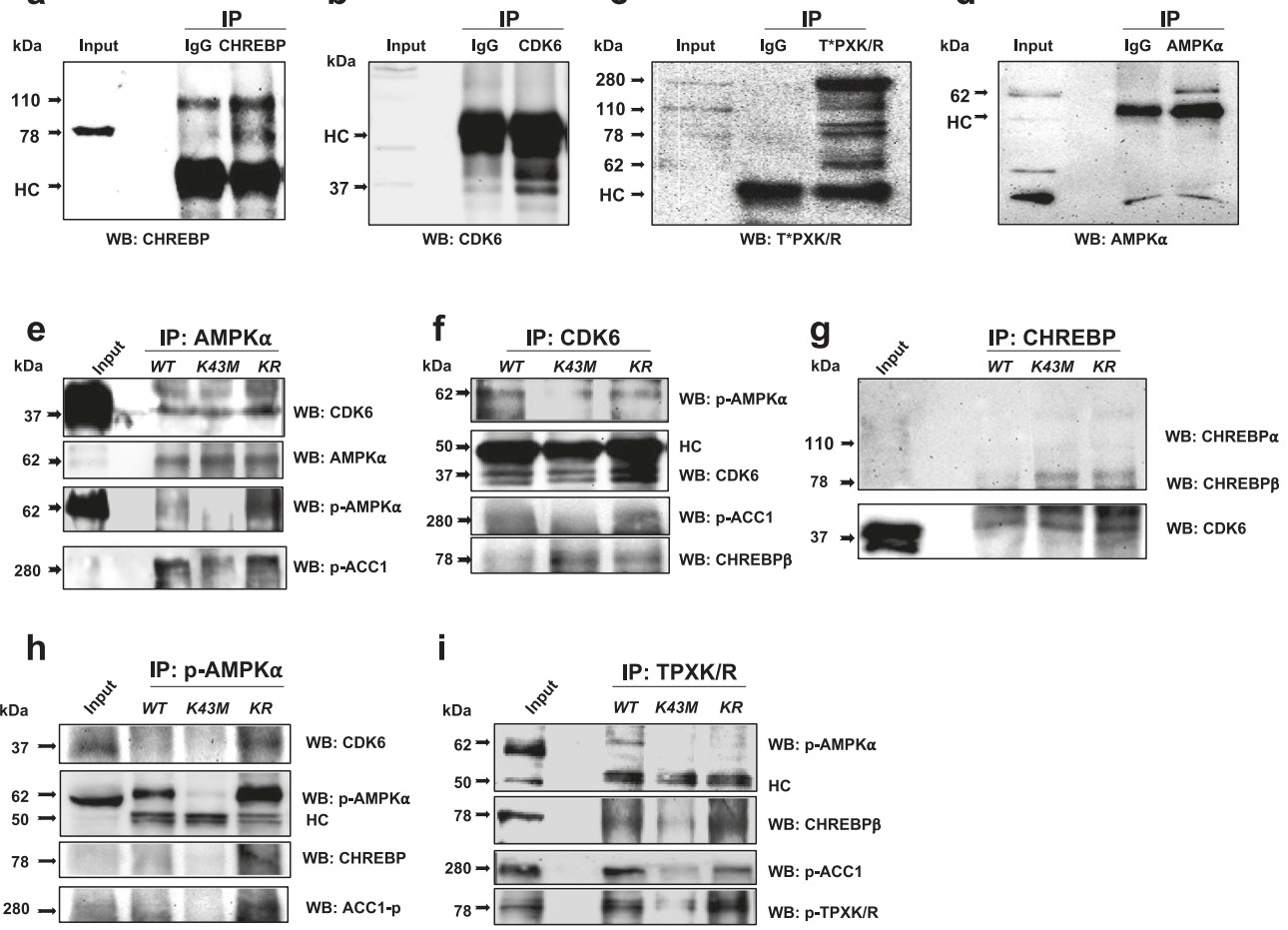

**Fig. 6 | CDK6 interacted with and phosphorylated AMPKα and CHREBP. a–d** IP-Western. The same amount (1 μg) of non-immune rabbit IgG and IgG of CHREBP **a**, CDK6 **b**, proteins containing CDK phosphorylated consensus motifs (p-T*PXK/R) **c**, and AMPKα **d** were used to immunoprecipitate eWAT extracts and blotted with the respective antibodies for CHREBP **a**, CDK6 **b**, T*PXK/R **c**, and AMPKα **d**. HC indicates a heavy chain of IgG. **e–i** IP-Westerns. AMPKα, CDK6, CHREBP, p-AMPKα, or proteins containing consensus CDK phosphorylation motif (p-T*PXK/R) was immunoprecipitated from eWAT extracts and blotted with the indicated antibodies. The Input represented 50 μg extracts from *WT* eWAT. Three-four independent experiments were repeated with similar results.

phosphorylation of S618 was only observed before reaction to CDK6/D3, suggesting an inhibitory effect of CDK6 on S618 phosphorylation. It is notable that AMPK-dependent phosphorylation site Ser 568, which has same motif (PxS*P) at (PES$^{568}$P)[46] as the putative CDK6 phosphorylation located on proline-rich domain important for DNA binding capacity of CHREBP, was not phosphorylated by CDK6. Similarly, the putative CDK6 phosphorylation motif T*PXR at (T$^{590}$PPR)[46] was not phosphorylated by CDK6.

We have also found the different phosphorylation sites on AMPK-α1 and AMPK-α2, although they have 90% amino acid sequence identity within the kinase catalytic domain (KCD, 1-314 for α1, 1-312 for α2) and 61% identity elsewhere[55]. As shown in Fig. 7 and supplementary Fig. 9, phosphorylation sites are spread out across the entire AMPK-α1 (Fig. 7b) but mainly located on the KCD and β/γ binding domains on AMPK-α2 (Fig. 7c). Of note, phosphorylation of Thr-172 of AMPK-α2 is considered as an absolute requirement for activation of AMPK[44], was observed upon CDK6/D3 reaction, suggesting CDK6 is a novel upstream kinase that activates AMPK in addition to LKB1[44].

Together, these results confirmed that CDK6 interacts with and phosphorylates human CHREBP, AMPK-α1, and AMPK-α2 directly.

### Nuclear abundance of CHREBPβ in *K43M* ADSCs
CDK6 phosphorylates AMPKα and CHREBP directly. Phosphorylation of CHREBP by CDK6 may then prevent CHREBP from entering the nucleus, resulting in the reduction of DNL in VAT, whereas K43M will have opposite effects on subcellular localization of CHREBP.

ADSCs was used to test this hypothesis. ADSCs are easier to obtain and amplify than mature adipocytes. Most importantly, ADSCs isolated from the stromal vascular fraction (SVF) of adipose tissue share some similarities to mature adipocytes. *K43M* ADSC have reduced phosphorylation level of AMPKα and increased protein level of CHREBPβ (Supplementary Fig. 10a, b). Moreover, CDK6 interacts with AMPKα/CHREBPβ (Supplementary Fig. 10c–g). Similar as mature adipocytes, *K43M* ADSCs had a reduction of CDK6-bound p-AMPKα, p-ACC1, and p-CHREBPβ (Supplementary Fig. 10c–g). Thus, ADSCs could be utilized as a surrogate stem cell model for mature adipocytes to study subcellular localization of CHREBP in the presence or absence of CDK6 kinase activity.

As phosphorylation of CHREBP by AMPKα prevents CHREBP from entering the nucleus[46], we used activation of AMPKα as a positive control. We first assessed the efficacy of activation of AMPKα by 5-aminoimidazole-4-carboxamide ribose (AICAR), a cell-permeable adenosine analog, in *WT* and *K43M* ADSCs in vitro. Upon entering the cells, it is phosphorylated by adenosine kinase to ZMP, an analog of AMP, which activates AMPKα by promoting its phosphorylation at T172 and by direct activation via an allosteric AMP site[56,57].

ADSCs were cultured in growth medium, and then switched to serum free medium for 16 h followed by treatment for 7 hours with 50

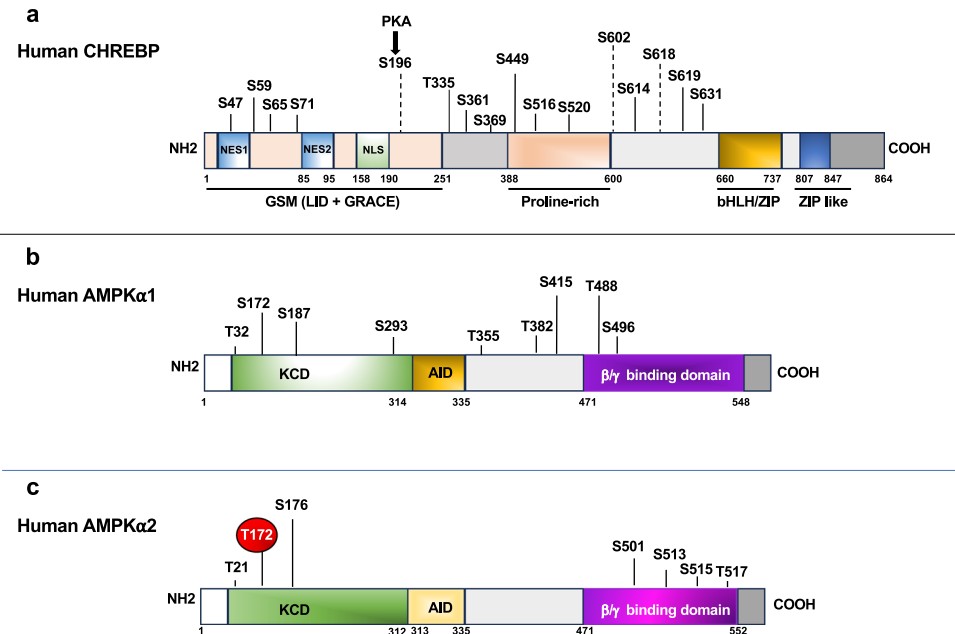

**Fig. 7 | CDK6 phosphorylated human AMPKα and CHREBP directly. a** Schematic depiction of phosphorylated sites on CHREBP, which contain a N-terminal domain (1-251), a proline-rich domain, and a DNA binding domain bHLH/ZIP. The N-terminal contains two nuclear export signals (NES1 and NES2) (1-95), a nuclear localization signal (NLS, 158-190), and domains named glucose-sensing module (GSM 1-251) critical for glucose regulation including low glucose inhibitory domain (LID) and a glucose-response activation conserved element (GRACE). The numbers above the schematic diagram indicated the phosphorylation occurred on serine (S)/threonine (T) residues. Non-CDK6-specific phosphorylation of S196, S602, and S618 were indicated by dotted lines. **b**, **c** Schematic depiction of phosphorylated sites on AMPKα1 **b** and AMPKα2 **c**, respectively. AMPKα contains a kinase catalytic domain (KCD, 1-314 for α1, 1-312 for α2), an auto inhibitory domain (AID), (313/315-335), and a β/γ binding domain (471-548/552). CDK6-specific phosphorylation sites were indicated by amino acid residue numbers above the schematic diagrams. Thr-172 was highlighted in red.

μM of AICAR, a condition known to be sufficient to activate AMPKα[58]. Activation of AMPKα was measured by immunoblotting with an antibody against p-AMPKα (T172). In agreement with the observations in eWAT (Fig. 5e, g) and ADSCs (Supplementary Fig. 10 a), in the absence of AICAR, the level of p-AMPKα and p-ACC1 in *K43M* ADSCs was reduced by 50% and 90%, respectively, as compared to those in *WT* ADSCs without AICAR (*WT*) (Fig. 8a and Supplementary Fig. 11 a, c). The level of total AMPKα was unchanged (Fig. 8a and Supplementary Fig. 11 b), but the level of total ACC1 was increased by 2.2-fold in the ADSCs of *K43M* mice as compared to that of *WT* (Fig. 8a and Supplementary Fig. 11 d), largely consistent with the observation in eWAT (Fig. 2g, i and Fig. 5e, g).

In the presence of AICAR, the level of p-AMPKα in *WT* and *K43M* ADSCs was increased by 50% and 40%, respectively, as compared to that in *WT* (Fig. 8a and Supplementary Fig. 11a). However, the levels of p-ACC1 in *WT* and *K43M* ADSCs were not significantly different from that of *WT* by AICAR treatment (Fig. 8a and Supplementary Fig. 11 c). These results indicate that *K43M* ADSCs had increased ACC1 protein expression level and comparable AMPKα to *WT* ADSCs but reduced level of p-AMPKα and p-ACC1, which can be re-activated to a level like that of *WT* ADSCs in the presence of AICAR. Thus, AMPKα could be activated in our system with AICAR[56,57].

Having assessed the efficacy of activation of AMPKα by AICAR, we next analyzed subcellular localization of CHREBPα and CHREBPβ proteins from *WT and K43M* ADSCs by immunoblotting after extracting nuclear and cytoplasmic fractions (Fig. 8b, c). CCAAT/enhancer-binding protein (C/EBP-β), which has different isoforms including C/EBPβ (full length, LAP*), C/EBPβ (LAP, activator), and C/EBPβ (LIP), was used as positive controls for nuclear fraction since wild-type C/EBPβ is found almost exclusively in the nucleus[59] (Fig. 8b), both in the absence or presence of AICAR. In the absence of AICAR, the level of nuclear CHREBPβ was increased by 1.8-fold in *K43M* ADSCs as compared to that in *WT* ADSCs (Fig. 8b and Supplementary Fig. 11 e), while the level

of cytoplasmic CHREBPβ was similar between *WT* and *K43M* ADSCs (Fig. 8c and Supplementary Fig. 11f). In the presence of AICAR, nuclear CHREBPβ was increased by 520-fold (2.6/0.005) in *K43M* ADSCs as compared to that in *WT* ADSCs (Fig. 8b and Supplementary Fig. 11e), while the cytoplasmic CHREBPβ was increased by only 1.5-fold (4.5/3.1) in *K43M* ADSCs as compared to that of *WT* ADSCs (Fig. 8c and Supplementary Fig. 11f). Thus, AICAR seems to particularly enrich nuclear CHREBPβ in *K43M* ADSCs. Of note, the levels of cytoplasmic CHREBPα were very similar between *WT* and *K43M* ADSCs both in the absence and presence of AICAR, whereas there were no detectable CHREBPα found in the nucleus (Fig. 8b and Supplementary Fig. 11g). Together, these data demonstrate that nuclear CHREBPβ was significantly increased in *K43M* ADSCs as compared to that in *WT* ADSCs, and that AICAR was able to prevent nuclear localization of CHREBPβ in *WT* ADSCs but not in *K43M* ADSCs, indicating a role of CDK6 in the AMPKα-CHREBPβ axis in regulating DNL.

## Knockdown of CHREBP in *K43M* cells reduces expression of SCD1 protein in vitro

To further validate the effects of CHREBP in K43M-induced DNL, we used shRNA to knockdown *CHREBP* in *K43M*-ADSC cells (Fig. 8d) and examined the resultant changes in the expression of DNL-related enzymes (Fig. 8e, f). *Chrebp*-specific (*ShRNA-CH*) or vector (V) transfected *K43M*-ADSC cells were selected by different concentrations of puromycin (2/4/6 ug/ml) for 5 days. Figure 8d shows CHREBP protein levels in the cells selected under 3 concentrations of puromycin, which suggests that 4 ug/ml of puromycin was optimal and was thus used in subsequent experiments.

Surprisingly, we only observed that SCD1 mRNA (Fig. 8e) and protein (Fig. 8f) were markedly downregulated, indicating SCD1 is transcriptionally regulated by CHREBP in vitro. However, other DNL-associated genes continued to be up-regulated or down-regulated marginally but significantly in cells lacking CHREBP (Fig. 8e). Similarly,

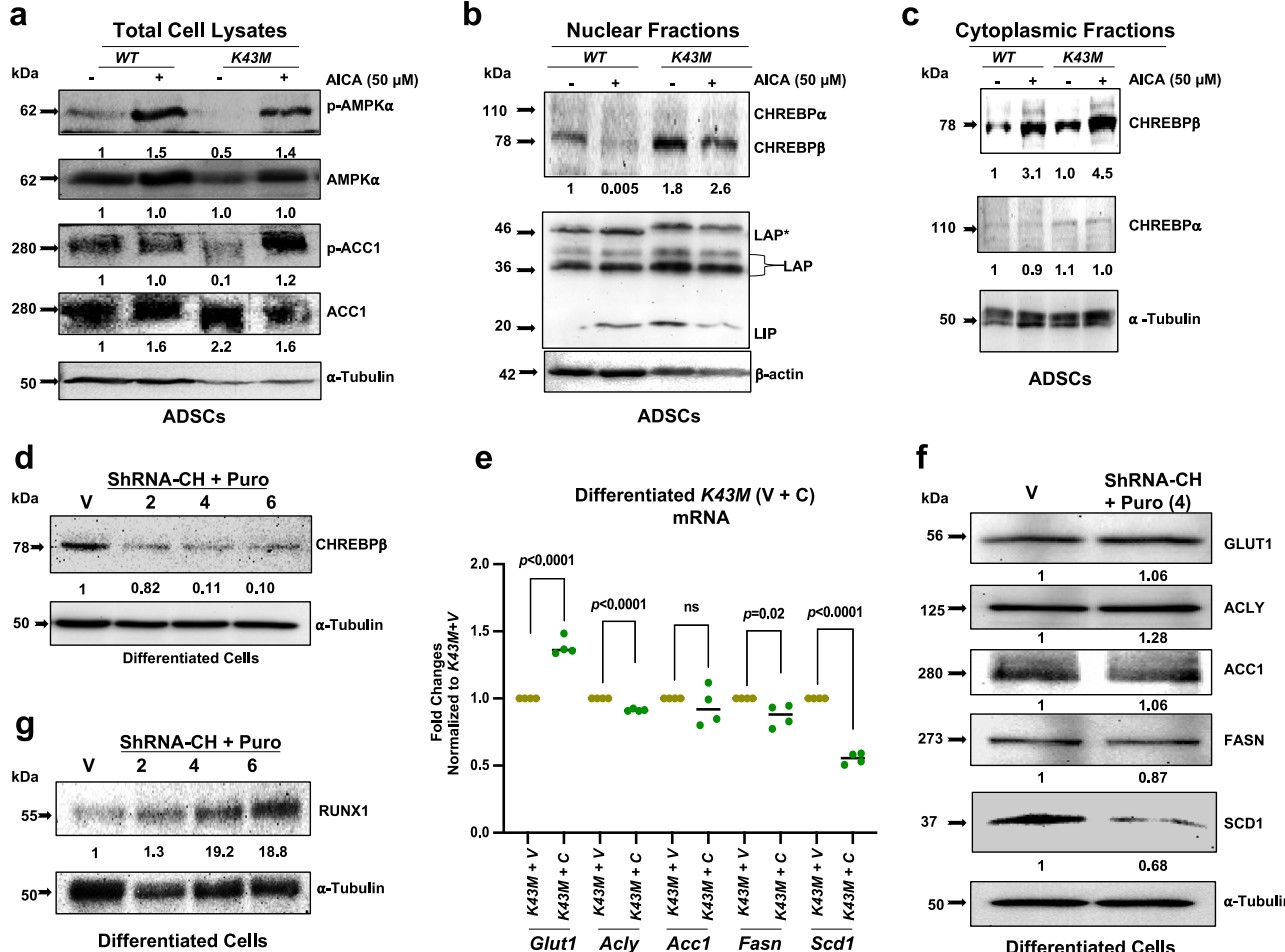

**Fig. 8 | Loss of CDK6 kinase activity leads to nuclear abundance of CHREBPβ. Loss of CHREBP results in reduction of SCD1.** a–c loss of CDK6 kinase activity renders nuclear localization of CHREBPβ. Total cell lysate **a**, nuclear **b**, and cytoplasmic **c** fractions were immunoblotted with different antibodies as indicated. α-tubulin or β-actin was used as loading control. The intensity of each protein in **a**–**d**, **f**, **g** was measured by **FluorChem M**, initially normalized to its internal control α-tubulin/β-actin, and subsequently normalized to the controls, which was arbitrarily set to 1 unit. **d**–**f** Effect of CHREBP knockdown on expression NDL-related genes/proteins. **d**, **g** immunoblots of CHREBPβ **d** and RUNX1 **g** in knockdown differentiated *K43M* cells selected by different concentration of puromycin.

**e**, **f** mRNA **e** and protein **f** levels of *Glut1*, and DNL-specific markers (*Acly, Acc1, Fasn, and Scd1)* in vector control (*K43M + v*) and in CHREBPβ knockdown (*K43M + C*) differentiated *K43M* cells. The fold changes of each mRNA were normalized to the (*K43M + v*) control, which was arbitrarily set to 1 unit. Data shown in **e** were mean ± SE (*n* = 5). We calculated statistical significance using two-tailed Student's T-test, with *p* < 0.05 considered significant. P values were displayed above two groups. NS indicates no significance between two groups. All the immunoblots shown are representative experiment chosen from 3-4 independent experiments.

we did not find notable changes in the protein levels, including GLUT1, ACLY, ACC1, and FASN (Fig. 8f), suggesting the compensation of loss of CHREBP by other transcription factor(s) including MondoA[60] and Srebp-1c[12]. Indeed, consistent with previously findings[61], RUNX1 protein was markedly upregulated in differentiated CHREBP-knockdown cells (Fig. 8g), which may also compensate the loss of CHREBP in *K43M* cells.

## Discussion

In the present study we demonstrated an essential role of CDK6 on negative regulation of DNL in VAT but not in the liver. *K43M* mice displayed increased DNL in VAT in vivo, accompanied by markedly increased GLUT1, CHREBP, and lipogenic enzymes in WAT but not in the liver. Inhibition of CDK6 kinase activity in *WT* mice under HFD by CDK6 inhibitor recapitulated the phenotypes observed in *K43M* mice. Mechanistically, we demonstrated that 1) phosphorylation of AMPKα on Thr-172 by CDK6 leads to phosphorylation and inactivation of ACC1; 2) phosphorylation of CHREBP by CDK6 prevents CHREBP from entering the nucleus; 3) ablation of RUNX1 in *K43M* mature adipocytes reverses most of the phenotypes observed in *K43M* mice; and 4) RUNX1 cooperates with CHREBP in regulation of DNL in VAT. Together with previous

studies[22], these data demonstrate that inhibition of CDK6 could increase DNL in WAT but not in the liver of obese individuals, enhance beige cell formation, and improve glucose tolerance and insulin sensitivity sensitivity[22]. Thus, inhibition of CDK6 kinase activity provides a potential therapy for obesity-related metabolic diseases.

The exact mechanism for the differential effect of CDK6 on DNL between VAT and the liver is unknown. One possibility could be the different signaling pathways involved in the two organs. For instance, phosphorylation of mTOR and AMPKα was reduced in adipose but increased in the livers (Fig. 5). This dichotomous regulation of mTOR and AMPKα phosphorylation could be an underlying reason for the opposite regulation of DNL in the VAT and the liver. Proteomic analyses revealed AMPKα and CHREBP as novel substrates for CDK6. There are putative CDK phosphorylation motif as T*PXR at $T^{590}$PPR[46] and as PXS*P at PPES$^{568}$P[46] in CHREBP. Similarly, human, or rodent AMPK-α1 and AMPK-α2 have putative CDK phosphorylation motif as T*PXR at $T^{490}$PQR. However, we did not detect the phosphorylation sites on those putative CDK phosphorylation motifs in vitro, which may be phosphorylated by other CDKs (Fig. 6i). Instead, we have identified multiple phosphorylation sites on CHREBP, AMPKα1 and

AMPKα2. Although the significant of CDK6-mediated phosphorylation awaits future investigation, phosphorylation on Thr[172] of AMPKα2 yielded breakthrough since the identification of an upstream kinase in the AMPK cascade is a long-awaited advance in the field, which may confer the kinase activity of AMPK in the absence of LKB1[62].

We have also found that some phosphorylation sites are located on amino acids 1-100 of CHREBP, which contains two nuclear export signals (NES) and domains named glucose-sensing module (GSM) critical for glucose regulation including LID and a glucose-response activation conserved element (GRACE), while others are dispersed on different domains. Thus, it is conceivable that in the presence of CDK6 kinase activity, phosphorylation on NES results in nuclear export of CHREBP, leading to reduced transcriptional activation of target genes; phosphorylation on GSM causes CHREBP to fail to regulate the glucose metabolism; phosphorylation on proline-rich domain engender loss of protein-protein interactions.

Consistent with in vitro phosphorylation analysis, CDK6 interacts with AMPKα and phosphorylate on T172 (Figs. 5, 6) in vivo, which is considered as an absolute requirement for activation of AMPK[44]. Loss of CDK6 kinase activity leads to reduced levels of p-AMPKα on T172 and p-ACC1 on S79 in WAT (Fig. 5) and reduced CDK6-bound p-AMPKα

and p-ACC1 (Fig. 6), whereas *K43M* liver had slightly but significantly increased levels of p-AMPKα and unchanged p-ACC1 (Fig. 5f, h). Phosphorylated AMPKα leads to phosphorylation of ACC1[47] (Fig. 5e, g), an important rate-limiting enzyme for the synthesis of malonyl-CoA. Hyper-phosphorylation of ACC1 translates into a reduction in lipid synthesis rates[47] (Fig. 1).

Consistent with in vitro phosphorylation analysis, CDK6 interacts with and phosphorylates CHREBP (Fig. 6) in vivo. Loss of CDK6 kinase activity leads to higher protein levels of CHREBP in the nucleus (Fig. 8b and Supplemental Fig. 11e). Thus, these data reveal an unappreciated post-transcriptional regulation of AMPKα and CHREBPβ by CDK6. Future studies are needed to elucidate how loss or inhibition of CDK6 kinase activity induces different signaling pathways in VAT and liver.

In addition to post-translational modification, DNL in *K43M* mice may be regulated transcriptionally. RUNX1 is a known target of CDK6[22,24,32]. Phosphorylation of RUNX1 by CDK6 and other kinases such as CDK1 and CDK2[24] promotes RUNX1 proteolytic degradation[24], resulting in a reduction of RUNX1 recruitment to the proximal promoter regions of *Pgc-1α and Ucp-1* genes[22], subsequently leading to reduced level of BAT-specific protein expression[22]. In contrast, phosphorylation of RUNX1 by extracellular signal-regulated kinases enhances its transcriptional activation[63]. Consistent with this notion, RUNX1 was upregulated in iWAT[22], eWAT, and eWAT-derived-ADSCs of *K43M* mice as compared to that of *WT* mice (Supplemental data Fig. 2e, g). Loss of RUNX1 (Fig. 2) partially (ACLY and FASN) or fully (ACC1, SCD1) reversed the increased protein levels observed in *K43M* eWAT (Fig. 2), suggesting that RUNX1 activity alone does not fully account for the stimulation of DNL, and that some other molecules or mechanisms are also involved in CDK6-mediated negative regulation of DNL.

It is known that MondoA[60] and Srebp-1c[12] act in synergy with CHREBP in the full induction of glycolytic and lipogenic gene expression in the liver[19]. Viral re-expression of CHREBP in the liver of SREBP-1c knockout mice normalized glycolytic gene expression but not lipogenic gene expression in spite of the well-known function of CHREBP in the control of DNL-related genes[19], indicating that some other molecules or mechanisms may have been induced to counteract the effect of over-expression of CHREBP. In fact, we found that knockdown of CHREBP did not causes a remarkable decrease in gene and protein expression associated with DNL. Consistent with previous report[61], an increase in RUNX1 was found in CHREBP-knockdown cells (Fig. 8g), which may compensate the loss of CHREBP on DNL, as manifested by unchanged expression of most of the DNL-associated protein including GLUT1, ACLY, ACC1, and FASN (Fig. 8e, f). In addition, supplement of the PPARγ ligand, rosiglitazone, in the culture medium may also play a partial role in rescuing the defect caused by CHREBP deficiency, including adipogenic marker genes expression and lipid accumulation[6], since PPARγ is the master transcription factor in adipogenesis[64]. Thus, CHREBP reversely correlated RUNX1.

Despite our current lack of understanding of the mechanistic links between RUNX1-mediated regulation of CHREBP, analysis of different set of metabolic profiles provides clues to the phenotypic responses observed in CDK6 mutant animals. For example, ablation of RUNX1 consistently reduced DNL in both *WT* and *K43M* mice, whereas ablation of RUNX1 resulted in reduction of CHREBP in *K43M* but not in *WT* mice, consistent with a previous study showing that conditional knockout of RUNX1 on *WT*-iWAT does not change white fat browning but reverse increased white fat browning in *K43M*-iWAT, indicating that the CHREBP expression in the presence of intact CDK6 is RUNX1-independent. Thus, Future in vivo studies are needed to elucidate if CHREBPβ and RUNX1 act independently or collaboratively mediate the regulatory function of CDK6 in DNL.

In summary, the current study suggests that CDK6 negatively regulates DNL in VAT but not in the liver by phosphorylating RUNX1[22,24,32], AMPKα, and CHREBP (Fig. 9). Giving the well-established role of DNL in obesity and other metabolic diseases, our findings will

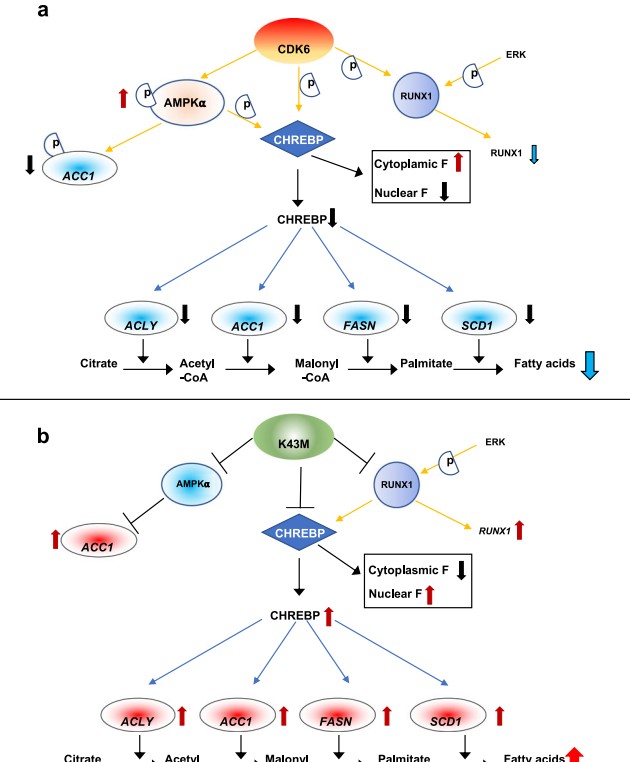

**Fig. 9 | A working model for the role of CDK6 in negative regulation of DNL by phosphorylating RUNX1, AMPK-α, and CHREBPβ. a** Phosphorylation of RUNX1 by CDK6 and other kinases such as CDK1 and CDK2[24] promotes RUNX1 proteolytic degradation[24], resulting in a reduction of RUNX1 recruitment to the proximal promoter regions of its target gens such as *SCD1*[33], subsequently leading to reduced level of DNL-specific protein expression. CDK6 phosphorylates and activates AMPKα, which in turn phosphorylates CHREBP and ACC1. CDK6 also phosphorylates CHREBPβ. Hyper-phosphorylation of CHREBP leads to accumulation of CHREBP in the cytoplasm and reduced transcription of target genes[46], and hyper-phosphorylation of ACC1[47], an important rate-limiting enzyme for synthesis of malonyl-CoA, which results in a reduction in lipid synthesis rates[47]. **b** In the absence of CDK6 protein/kinase activity, RUNX1 is stabilized, DNL-specific protein expression is increased, and hypo-phosphorylation of CHREBP leads to accumulation of CHREBP in the nucleus and increased transcription of target genes[46], and hypo-phosphorylation of ACC1[47] translates into an increase in lipid synthesis rates[47].

hopefully stimulate development of novel strategies aiming at blocking CDK6 kinase activity or its downstream effectors in the therapy of obesity-related diseases, such as T2D.

## Methods

### Mice

**Generation of different mature adipocytes-specific Runx1$^{-/-}$ and K43M;Runx1$^{-/-}$ mice.** Mature adipocyte-specific *Runx1$^{-/-}$* mice, and *K43M;Runx1$^{-/-}$* were produced by crossing *Runx1$^{fl/fl}$* (Jackson lab, stock number: 010673) and *K43M;Runx1$^{fl/fl}$* mice with Adipoq-Cre mice (Jackson lab, stock number: 010803), respectively. All experiments were performed according to the guidelines of the Institutional Animal Care and Use Committee of Tufts University.

### In vivo DNL (UMass Chan Medical School- Metabolic Disease Research Center)

Male and female 6-8-month-old *WT*, *K43M*, and *KR* mice under NCD were intravenously injected with 10 µCi of D-[$^{14}$C(U)]-glucose. Blood samples were collected at 0, 5, 10, 30 and 60 min for measurement of D-(U-$^{14}$C)-glucose concentrations to assess the isotopic enrichment of circulating glucose and the systemic clearance of D-(U-$^{14}$C)-glucose. At the end of experiment, e/gWAT or livers were isolated from mice and DNL was assessed by measuring D-(U-$^{14}$C)-glucose incorporation into organ-specific TG. Briefly, tissues were homogenized with chloroform/methanol, and lipid layer was extracted using H$_2$SO$_4$. Organ-specific TG levels were measured via colorimetric assay, and [$^{14}$C]-labeled TG was measured using liquid scintillation. The ratio of [$^{14}$C]-labeled TG in total organ-specific TG levels were calculated.

### In vitro DNL

The primary ADSCs from the SVF of e/gWAT of *WT* or *K43M* mice were isolated, as previously described[22,65]. MEFs (*WT, Cdk4-KO*)[39] were obtained from Dr. Sicinski of Dana Farber Cancer Institute.

To induce differentiation, adipocyte stem/progenitor cells were cultured for another 48 hours after reaching confluence, then induced with DMEM/F12 medium containing FBS (10%), insulin (0.5 µM), dexamethasone (100 nM), 1-methyl-3-isobutylxanthine (IBMX, 500 µM) and rosiglitazone (1.0 µM) for 7 days. On day 8, accumulation of lipid-containing cells was detected by Oil red O staining as described[66]. Fluorescent photomicrographs of differentiated cells were obtained. Red fluorescence indicates the Oil-Red-O staining. In a subset of experiments, the differentiated cells were also used for RT-PCR and immunoblotting experiments.

**For knockdown of CHREBP.** PLKO.1 target gene was purchased from Sigma Aldrich (Cat# TRCN00000234183). The clone contains target sequence of TGT CAT CCT GGA GGG TAA TTA, which has been vilified by the manufacture to be able to knockdown 84% of protein expression.

293 T cells were cultured in high-glucose DMEM supplemented with 10% FBS. Recombinant virus plasmid PLK0.1-ShRNA-CHREBP (ShRNA-CH) (Sigma-Aldrich, SHCLNG) and packaging plasmids (Δ8.2 and VSVG) were co-transfected into 293 T cells using Lipofectamine 2000 (Invitrogen). After 48 h and 72 h, the supernatant containing lentiviral particles were harvested and filtered through 0.45 µm filter. The concentrated lentiviral solution was then incubated with 80% confluent *K43M*-ADSCs in DMEM/F12 medium containing 15% FBS supplemented with polybrene (8 µg/ml, American Bioanalytical) for 24 h, then changed to fresh 15% DMEM/F12. The infected *K43M*-ADSCs were selected by puromycin (2/4/6 µg/ml) for 5 days to determine the optimal puromycin concentration, which is 4 µg/ml.

### Lipid content tests

The concentrations of FFA, TG, and cholesterol in serum and liver were measured by using non-esterified fatty acid test (Wako Pure Chemical, Osaka, Sigma), INFINITY Triglycerides Reagent (Sigma), and INFINITY Cholesterol Reagent (Sigma), respectively, according to the manufacturer's instructions.

### Immunoblotting, and IP-Western

Cell lysates were prepared as described[28]. Western blotting, and IP-Westerns were performed as described[28]. The intensity of each protein was determined by FluorChem M system and normalized to α-tubulin. The fold change of each protein was normalized to the relative controls, which was arbitrarily set to 1 unit.

Antibodies used in this study included CDK6 (C-21, Santa Cruz), RUNX1 (Ab23980, Rabbit polyclonal antibody, Abcam), α-Tubulin (Sigma, T6199), GLUT1 (12939 S, Cell Signaling Technology-CST), GLUT4 (2213 S, CST), ACLY (4332 S, CST), FASN (8335 T, CST), p-ACC1 (3661 S, CST, S79), ACC1 (3676 S, CST), SCD1 (2438 S, CST), p-AMPKα (2535 S, CST, T172), AMPKα (2603 S, CST), CHREBP (58069 S, CST), mTOR (2983 S, CST), p-mTOR S2448 (5536 S, CST), p-PXS*P (2325 S, CST), and p-T*PXK/R (14371 S, CST).

### Quantitative real-time PCR

The experimental procedures were same as those described previously[22,67]. The 36B4 gene, encoding an acidic ribosomal phosphoprotein P0 (RPLP0), was used as an internal control. The Primer sequences are listed in Supplemental Table 1.

### Drug treatment

For in vivo treatment with Lee/V, HFD was started at the age of 4 weeks. Lee (200 mg/kg daily)[32] or V (0.5% methycellulose) was administered by gavage at the age of 6 weeks for 14 weeks.

### Fusion proteins (CHREBP, AMPK-α1, and AMPK-α2)

Human ORF Clones were obtained from OriGene Technologies Inc. 293 T cells were transfected with 500 ng of PCMV6-entry vectors (PCMV6-entry-C-Myc-DDK vector) encoding AMPK-α1 (RC218572), AMPK-α2 (RC210226), and CHREBP (RC220626) for 72 hours. The cells were harvested and whole cell lysates were immunoprecipitated with magnetic beads conjugated anti-Flag (DDK) antibody (Sigma Anti-Flag M2 Magnetic Beads, M8832), which recognizing Myc-DDK-tagged fusion proteins and followed by purification with magnetic separator.

### In vitro CDK6/CyclinD3 kinase assay

The purified fusion proteins were divided into two parts, one part was used as the substrates for kinase assay with active CDK6/Cyclin D3 kinase (EMD Millipore Corporation, 14-519) in the presence of 100 µM ATP at 30 °C for 20 min. The reaction was stopped by adding 2x sample buffer and the mixture was separated by SDS-PAGE. The gels were stained with Coomassie Brilliant Blue, and visibly stained proteins (~1 µg) were excised and submitted to proteomic analysis at Mass Spectrometry Core of Beth Israel Deaconess Medical Center. The other part was directly separated on the SDS-PAGE, stained with Coomassie Blue, excised, and subjected to proteomic analysis as non-phosphorylation control.

### Statistical methods and analysis

For most experiments, the sample size was chosen based on expected differences between experimental and control groups to provide adequate power to detect a significant difference specifying α = 0.05, two-tailed testing, and power (= 1-β) of 80%, using commercially available software packages (Statistical Solutions nQuery Advisor; http://www.statsol.ie/nquery/nquery.htm). All data are expressed as means ± S.E. We calculated statistical significance using two-tailed Student's T-test, with $p < 0.05$ considered significant. For Fig. 4a and Supplementary Fig. 3a–d, we used different numbers of * for each comparison. *$p < 0.05$, **$p < 0.01$, ***$p < 0.001$, ****$p < 0.0001$, significantly different from its control. NS indicates no significance

between two groups. The exact *p* values were presented in source data files. For the rest of figures, the *p* values were summarized above two compared groups. NS indicates no significance between two groups.

### Reporting summary

Further information on research design is available in the Nature Portfolio Reporting Summary linked to this article.

## Data availability

The data that support the findings of this study are available from the corresponding author upon request. The Mass Spectrometry data including raw and peak data that support the findings of this study are available in MassIVE with the reference number MSV000093063, and the result data generated by Dr. John M Asara from BIDMC are available at Figshare, https://doi.org/10.6084/m9.figshare.24265774. Source data are provided in this paper.

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

## Acknowledgements
This work was supported by DOD (W81XWH1910301), a Tufts CTSI-Catalyst Awards (UL1 TR002544), and AG078484 to M.G.H.; DK117163 and DK134534 to S.R.F.; P01CA120964 and R35CA197459 to J.M.A. We thank Dr. Peter Sicinski (Dana Farber Cancer Institute, Boston, MA USA) for providing MEFs (*WT, Cdk4-KO*).

## Author contributions
A.J.H and W.L designed and performed experiments, analyzed data, and interpreted results; C.D, Y.Z, J.K.H, X.H, Z.Y assisted in genotyping and performed experiments; J.M.A assisted in designing and performing proteomic analysis; G.F.H helped in proteomic analysis; S.G.D assisted in writing and editing; S.R.F provided scientific and technical guidance; and M.G.H. designed experiments, interpreted results, provided guidance for the group, and wrote the manuscript.

## Competing interests
The authors declare no competing interests.
