## [Peer Review File · Nature Communications]

CDK6 Inhibits De Novo Lipogenesis in White Adipose Tissues but not in the LiverREVIEWER COMMENTS

Reviewer #1 (Remarks to the Author):

The participation of CDK6 in metabolic control has been previously described by Dr. Hu laboratory and others. In this manuscript, the authors go one step further and analyze the role of CDK6 in the regulation of de novo lipid synthesis (DNL). They show increased lipogenesis in a mouse model expressing a CDK6 kinase-dead mutant. Interestingly, they show that this is an adipose-tissue restricted effect, which is not observed in the liver. By using another mouse model that does not express RunX1, they propose that the effects of CDK6 are mediated by this protein. They further propose that CDK6 phosphorylates CHREBP and AMPK, ultimately resulting in the inhibition of lipogenesis. Consequently, mouse deficient in CDK6 activity has decreased DNL. The concept that CDK6 regulates this process in WAT, but not in the liver is novel. However, the general conclusions are not supported by the results. Specifically, this reviewer has the following concerns and suggestions.

1. The results section requires a detailed description of the mouse models that are used in the study. What are the K43M mice or the RunX1 mice? The reader must read other publications of the group to understand the models.
2. In vivo DNL show that TGs incorporate more carbon from glucose in the K43M model. Since the authors claim that this effect is specific for adipose tissue, the same experiment must be performed in liver.
3. The authors show that deletion of RunX1 in WT mice does not change the expression of CHREBP but decreases the expression of DNL proteins. In contrast, in the KR mouse model, deletion of RunX1 abrogates the effects of the mutant CDK6 on the expression of CHREBP and DNL genes and protein. It cannot be concluded from these results that RunX1 regulates the expression of CHREBP. Why the expression of DNL genes is decreased in RunX1 KO mice and not the expression of CHREBP?
4. Experiments under HFD both in WAT and liver are confusing. Indeed, it is known that HFD decreases DNL in normal mice. Lipogenesis is typically increased in response to a high-carbohydrate diet both in liver and in adipose tissues. Glucose metabolites generated during glycolysis activate ChREBP, whereas HFD inhibits DNL in adipocytes through blocking the activation of ChREBP- β (Nat Commun. 2016; 7: 11365.). It is surprising that in this manuscript the authors show increased expression of DNL genes under HFD. Feeding mice with a high sucrose diet would better reflect changes in DNL in both WAT and liver.

5. The results presented in the figure 3b are surprising. There is no detectable expression of SCD1 in the liver, which is typically highly expressed in this tissue. Moreover, the expression of tubulin is very heterogeneous, which make difficult the normalization of the results.

6. Treatment of mice with CDK4/6 inhibitors results in decreased fat mass, and increased expression of lipogenic genes. There are three questions concerning these results. First, the effects on adipose tissue of CDK4 inhibitors have been previously reported (JCI Insight 2018 Sep 6;3[17]:e123000 doi: 10.1172/jci.insight.123000). Second, the authors cannot conclude that the observed effects are secondary to CDK6 inhibition, since ribociclib also inhibits CDK4. Indeed, several articles support a role of CDK4 in adipose tissue biology. CDK4^{-/-} mice are adipose tissue deficient, and hyperactive CDK4 mice (R24C) have increased fat mass. And third, the results do not prove that the decreased fat mass in treated mice is the result of decreased DNL. The authors need to address these questions.

7. The regulation of mTOR is opposite in WAT and in liver. The authors could further explore mTOR activity in these tissues. Some studies have suggested that DNL in adipose tissue requires the mTORC2 pathway (Nature Communications volume 7, 11365 [2016] . An explanation is, at least, required of why mTOR is decreased in WAT and increased in liver without any consequences in lipid synthesis.

8. In figure 6 the western blots are of bad quality. The controls are missing [IP mock] to prove the specificity of the bands. The results are not properly interpreted. No differences are observed in CDK6 between the WT and K43M in the AMPK immunoprecipitates [6a], contrary of what the authors claim in the text. In the WB of p-AMPK in the CDK6 immunoprecipitates [6b] there is a problem with the K43M lane [probably a transfer issue].

9. The authors claim that CDK6 phosphorylates both AMPK α and CHREBP. They fundament this hypothesis by performing immunoprecipitation experiments. These experiments do not prove that these proteins are direct phosphorylation targets of CDK6. This is an interaction assay. Phosphoproteomic analysis and in vitro phosphorylation assays are required to suggest that AMPK and CHREBP are CDK6 targets. Moreover, in the model that the authors use, CDK6 is expressed, although it does not have kinase activity. I would assume that this does not preclude the interaction with its targets. A control of cyclins D could be used.

10. The authors could use some cell models [primary adipocytes] to test some of their hypotheses and mechanisms. They could inhibit CHREBP [siRNA or CrispR] or reintroduce AMPK mutants in these models and test DNL. They could also inhibit CDK4 to prove the relative contribution to the phenotype.

11. The authors do not provide mechanisms explaining why CDK6 regulates DNL in WAT but not in liver. This is one of the major findings of this study and merits further work.

Minor

12. In figure 1 d-h, the Y-axis scale should start at 0. Otherwise, the graph does not reflect the real amplitude of the differences.

13. In figure 2d, the quantification of the western blot of CHREBPb does not reflect the image of 2b.

14. Expression of CHREBPs should be shown in response to HFD.

15. In fig 3 c-d the results are represented as fold changes. However, it is not indicated relative to what. For instance, WT has a value of 0.5, or other values, but not 1. The same is true for other graphs in other figures.

Reviewer #2 (Remarks to the Author):

In this manuscript, the authors hypothesize using Cdk6 knock out mice, and Cdk6 kinase dead or K43M mice, and Runx1 knockout mice, that tonic expression of Cdk6 suppresses de novo lipogenesis in white adipose tissue, but not liver, via the phosphorylation of AMP kinase and ChREBPbeta. This is a follow-up from a previous study showing that Cdk6 suppresses Runx1 activity, thus limiting beiging of white adipose tissue. The authors used metabolic labelling of de novo lipogenesis to demonstrate increased de novo lipogenesis in white adipose tissue but not liver. Mechanistically, they found that Cdk6 phosphorylates and activates AMP kinase, and phosphorylates and inactivates ChREBPbeta. The general idea is that increased adipose lipogenesis leads to healthy storage of Tgs in adipose and this leads to insulin sensitivity, whereas Tg storage in liver leads to steatosis and insulin resistance. Thus, inhibition of Cdk6 may be therapeutic for obesity and diabetes. Indeed, systemic treatment with a Cdk6 inhibitor recapitulated the phenotype of the loss of Cdk6 function animals.

The data generated are clean and the study is generally well done. However, there are a number of issues that need to be addressed. One is that the antibody used here for ChREBPbeta has never been properly validated using overexpression or knockdown approaches and this should be done as many in the field find that ChREBPbeta turns over so rapidly that it is impossible to see in western blots. In addition, there are a few claims that seem overstated: 1) On page 5, the authors claim that "Loss of Runx1 fully reversed the phenotypes observed in K43M tissues, indicating that RUNX1 is the major downstream effector of CDK6 in DNL." This is not necessarily the case - Runx1 mutation is a suppressor of Cdk6 loss of function, which could be due to many possible indirect effects. 2) "RUNX1 is a major

downstream effector of K43M in DNL.” This should be re-written, as K43M is not a natural effect, therefore nothing can be downstream of it.

In addition, it is not clear to the readers why the authors use Runx1 or why this would reverse the phenotype of Cdk6 loss of function – this should be explained in the introduction or as an introduction to the results presented in the results section.

The numbering for mouse ChREBP is off by 2 base pairs – the numbering in the manuscript is for rat. Also, the sequence in human is slightly different in this region – this should be discussed. The phosphor-acceptor site for ChREBPbeta, S568, is also the acceptor site for AMP kinase, which has the same phenotype as the putative Cdk6 phosphorylation. Since the authors also see an increase in AMP kinase activity, the authors should design experiments to distinguish between the possibilities of Cdk6 phosphorylation and AMP kinase phosphorylation with in vitro assays.

In addition, there are a number of minor issues that should be addressed:

- 1) The molecular weights of the proteins should be marked in all the western blots
- 2) Are there any Runx1 binding sites in the ChREBP promoters?
- 3) The first paragraph of the introduction needs references.

Reviewer #1 (Remarks to the Author):

The participation of CDK6 in metabolic control has been previously described by Dr. Hu laboratory and others. In this manuscript, the authors go one step further and analyze the role of CDK6 in the regulation of de novo lipid synthesis (DNL). They show increased lipogenesis in a mouse model expressing a CDK6 kinase-dead mutant. Interestingly, they show that this is an adipose-tissue restricted effect, which is not observed in the liver. By using another mouse model that does not express RunX1, they propose that the effects of CDK6 are mediated by this protein. They further propose that CDK6 phosphorylates CHREBP and AMPK, ultimately resulting in the inhibition of lipogenesis. Consequently, mouse deficient in CDK6 activity has decreased DLN. The concept that CDK6 regulates this process in WAT, but not in the liver is novel. However, the general conclusions are not supported by the results. Specifically, this reviewer has the following concerns and suggestions.

1. The results section requires a detailed description of the mouse models that are used in the study. What are the K43M mice or the RunX1 mice? The reader must read other publications of the group to understand the models.

The description of our mouse models is now placed into introduction (page 3, lines 86-89).

K43M mice are CDK6 knock-in mice which has no kinase activity. The description of Runx1 mice is on page 4, lines 98-99.

2. In vivo DNL show that TGs incorporate more carbon from glucose in the K43M model. Since the authors claim that this effect is specific for adipose tissue, the same experiment must be performed in liver.

The data related in vivo DNL in liver is now included as **Supplementary Fig. 4**. The description about the data is on page 5, lines 119-126.

3. The authors show that deletion of RunX1 in WT mice does not change the expression of CHREBP but decreases the expression of DNL proteins. In contrast, in the KR mouse model, deletion of RunX1 abrogates the effects of the mutant CDK6 on the expression of CHREBP and DNL genes and protein. It cannot be concluded from these results that RunX1 regulates the expression of CHREBP. Why the expression of DNL genes is decreased in RunX1 KO mice and not the expression of CHREBP?

We agree with the reviewer and have discussed the potential relationship between CDK6, RUNX1, CHREBP and DNL genes (page 17-18, lines 433-454). It is obvious that RUNX1 plays content dependent roles on different conditions. We have observed similar phenotypes in our previous studies¹. For instance, ablation of RUNX1 on *WT* mice did not change body weight, white fat browning, energy expenditure, and beige cell related gene expression. However, ablation of RUNX1 on *K43M* mice reversed all the phenotypes observed in *K43M* mice. In this study, ablation of RUNX1 results in reduction of DNL-associated proteins in *WT*/*K43M*-VAT. However, we observed reduction of CHREBP in *K43M*-VAT but not in *WT*-VAT.

4. Experiments under HFD both in WAT and liver are confusing. Indeed, it is known that HFD decreases DNL in normal mice. Lipogenesis is typically increased in response to a high-carbohydrate diet both in liver and in adipose tissues. Glucose metabolites generated during glycolysis activate ChREBP, whereas HFD inhibits DNL in adipocytes through blocking the

activation of ChREBP- β (Nat Commun. 2016; 7: 11365.). It is surprising that in this manuscript the authors show increased expression of DNL genes under HFD. Feeding mice with a high sucrose diet would better reflect changes in DNL in both WAT and liver.

Consistent with previous studies, upon HFD feeding in *WT* mice, the DNL-related proteins, including ACC1, ACLY, and FASN are reduced, while SCD1 is upregulated which is consistent with phenomenon observed by other groups^{2,3}. What we emphasized in this manuscript is that the DNL-related proteins were increased significantly in *K43M* mice compared to those *in WT* mice under HFD (Fig.3 a. c), which is Glucose-independent.

In the revised manuscript, these data were presented in supplementary Fig. 7 and discussed on page 7, lines 168-173, 177-180.

5. The results presented in the figure 3b are surprising. There is no detectable expression of SCD1 in the liver, which is typically highly expressed in this tissue. Moreover, the expression of tubulin is very heterogeneous, which make difficult the normalization of the results.

We have repeated the experiments using β -actin as the control, which is more even than tubulin. Indeed, SCD1 was detectable from liver extracts of *WT/K43M/KR* mice under HFD. The new data are presented in Figure 3 b and d.

6. Treatment of mice with CDK4/6 inhibitors results in decreased fat mass, and increased expression of lipogenic genes. There are three questions concerning these results. First, the effects on adipose tissue of CDK4 inhibitors have been previously reported (JCI Insight 2018 Sep 6;3[17]:e123000 doi: 10.1172/jci.insight.123000). Second, the authors cannot conclude that the observed effects are secondary to CDK6 inhibition, since ribociclib also inhibits CDK4. Indeed, several articles support a role of CDK4 in adipose tissue biology. CDK4^{-/-} mice are adipose tissue deficient, and hyperactive CDK4 mice (R24C) have increased fat mass. And third, the results do not prove that the decreased fat mass in treated mice is the result of decreased DNL. The authors need to address these questions.

Response to question1: Yes, we cannot conclude that the observed effects are secondary to CDK6 inhibition, since ribociclib (LEE011) inhibits both CDK6 and CDK4, the latter of which has been also shown to be involved in HFD-induced obesity⁴ by positive regulation of lipogenesis and negative modulation of lipolysis⁵ as the reviewer 1 mentioned. However, the paper (JCI Insight 2018) about regulation of lipogenesis was retracted in 2023 due to the use of wrong animal models⁶.

To determine if CDK4 has a similar role as CDK6, we have performed DNL analysis *in vitro* by using *WT* and *CDK4*-deficient- MEFs and found that loss of CDK4 also increased expression levels of DNL-related genes including GLUT1, ACLY, ACC1, FASN, and SCD1. Thus, CDK4 has a similar role as CDK6 in negative regulation of DNL *in vitro*. The related description is on page 8, lines 196-203, in the revised manuscript. The related data is presented in supplementary Fig. 8.

Response to question 2: We agree with the reviewer on this. Based on our current observation, we cannot conclude that the decreased fat mass in treated mice is a result of increased DNL. We have pointed this out on Page 3 line 65-66 and have included to references to show that restoring DNL in WAT selectively reverts obesity-dependent insulin resistance (IR)^{7,8}.

Response to question3: Thus far, in our CDK6 mutant mouse models, the reduced fat mass in *K43M* mice are due to multiple effects of CDK6, including (1) increased white fat browning and energy expenditure¹, (2) reduced stem/progenitors (DOI 10.3389/fmolb.2023.1146047); (3) reduced capability of stem/progenitors to differentiate into white adipocytes (DOI10.3389/fmolb.2023.1146047), and (4) the negative regulation of DNL in WAT.

7. The regulation of mTOR is opposite in WAT and in liver. The authors could further explore mTOR activity in these tissues. Some studies have suggested that DNL in adipose tissue requires the mTORC2 pathway (Nature Communications volume 7, 11365 [2016]. An explanation is, at least, required of why mTOR is decreased in WAT and increased in liver without any consequences in lipid synthesis.

mTORC2 is required for DNL in adipose tissues (WAT and BAT) in part by controlling glucose uptake, which promote *ChREBPβ* expression. However, mTOR seems not required for *K43M*-mediated increase of DNL, since the levels of phosphorylated mTOR on S2448 (activation site) and total protein are reduced in *K43M* WAT. We have discussed these aspects on Page 8, Lines 205-212).

We will pursue the role of mTOR in DNL in WAT and liver in future studies to answer the question why the phosphorylation of mTOR S2448 and the total level of mTOR are regulated differently in WAT and liver, and what is the role of mTOR in DNL in the absence of CDK6 kinase activity.

8. In figure 6 the western blots are of bad quality. The controls are missing [IP mock] to prove the specificity of the bands. The results are not properly interpreted. No differences are observed in CDK6 between the WT and *K43M* in the AMPK immunoprecipitates [6a], contrary of what the authors claim in the text. In the WB of p-AMPK in the CDK6 immunoprecipitates [6b] there is a problem with the *K43M* lane [probably a transfer issue].

We have repeated the experiments with non-immune IgG as the control to sure the specificity. The new figures are presented as Figure 6 a-d. The description about the specificity of the bands is illustrated on page 9, lines 227-235.

9. The authors claim that CDK6 phosphorylates both AMPKα and CHREBP. They fundament this hypothesis by performing immunoprecipitation experiments. These experiments do not prove that these proteins are direct phosphorylation targets of CDK6. This is an interaction assay. Phosphoproteomic analysis and in vitro phosphorylation assays are required to suggest that AMPK and CHREBP are CDK6 targets. Moreover, in the model that the authors use, CDK6 is expressed, although it does not have kinase activity. I would assume that this does not preclude the interaction with its targets. A control of cyclins D could be used.

Following the reviewer's suggestion, we have performed proteomic analysis. The Mass Spectrometer data that support the findings of this study are available in MassIVE with the identifier (doi:10.25345/C57D2QJ4C) (reference number: MassIVE MSV000093063. The new figures are presented in Figure 7 and supplementary Figures 9 and 10. The description about the analyses and results are illustrated on page 11, lines 269-306. The significance of the phosphorylation sites is indicated on page 15-16, line 391-421.

The interaction of *K43M* and Cyclin D3 was published⁹. Co-IP experiments show the ability of *K43M* to p18, p27 and D cyclins was like that of WT-Δ (Figure 1),

Fig. 1. Interaction of K43M with D3, P18, and p27. IP-Westerns. CDK6 or Cyclin D3 was immunoprecipitated from thymocyte extracts and blotted with the indicated antibodies, HC indicates IgG heavy chain. Lane 1 (WT-Δ) represents input (50μg).

Hu MG et, al Blood, 2011; 117(23): 6120–6131

10. The authors could use some cell models [primary adipocytes] to test some of their hypotheses and mechanisms. They could inhibit CHREBP [siRNA or CrispR] or reintroduce AMPK mutants in these models and test DNL. They could also inhibit CDK4 to prove the relative contribution to the phenotype.

As suggested by the reviewer, we have used siRNA method to validate the effects of CHREBP in K43M-induced DNL. We ablated CHREBP with shRNAs targeting ChREBP (*ShRNA-ch*) in K43M-ADSC cells and then differentiated those cells. The new figures are presented in **Fig. 8 d-g**. The description about the analyses and results are illustrated on page 14-15, lines 362-376. The interpretation and discussion of results is on page 17-18, line 433-454.

11. The authors do not provide mechanisms explaining why CDK6 regulates DNL in WAT but not in liver. This is one of the major findings of this study and merits further work.

We have provided the mechanisms detailed in Figure 5, 6, 7, 8, and 9 and their related supplementary data.

Minor

12. In figure 1 d-h, the Y-axis scale should start at 0. Otherwise, the graph does not reflect the real amplitude of the differences.

The figure 1d-h has been re-organized.

13. In figure 2d, the quantification of the western blot of CHREBPb does not reflect the image of 2b.

Thanks for the comments. We have replaced more meaningful representative Figure 2b now.

14. Expression of CHREBPs should be shown in response to HFD.

These data are now shown in **Supplementary Fig. 7c**.

15. In fig 3 c-d the results are represented as fold changes. However, it is not indicated relative to what. For instance, WT has a value of 0.5, or other values, but not 1. The same is true for other graphs in other figures.

In Figure 3 c and d of the revised manuscript, “fold changes” have been replaced by Average of Intensity, which is presented using average of band intensity from three different samples derived from each type of genotype. α -tubulin was used as loading control. The intensity of each protein was measured by FluorChem M system and normalized to α -tubulin.

Reviewer #2 (Remarks to the Author):

In this manuscript, the authors hypothesize using Cdk6 knock out mice, and Cdk6 kinase dead or K43M mice, and Runx1 knockout mice, that tonic expression of Cdk6 suppresses de novo lipogenesis in white adipose tissue, but not liver, via the phosphorylation of AMP kinase and ChREBPbeta. This is a follow-up from a previous study showing that Cdk6 suppresses Runx1 activity, thus limiting being of white adipose tissue. The authors used metabolic labelling of de novo lipogenesis to demonstrate increased de novo lipogenesis in white adipose tissue but not liver. Mechanistically, they found that Cdk6 phosphorylates and activates AMP kinase, and phosphorylates and inactivates ChREBPbeta. The general idea is that increased adipose lipogenesis leads to healthy storage of Tgs in adipose and this leads to insulin sensitivity, whereas Tg storage in liver leads to steatosis and insulin resistance. Thus, inhibition of Cdk6 may be therapeutic for obesity and diabetes. Indeed, systemic treatment with a Cdk6 inhibitor recapitulated the phenotype of the loss of Cdk6 function animals. The data generated are clean and the study is generally well done.

However, there are a number of issues that need to be addressed. One is that the antibody used here for ChREBPbeta has never been properly validated using overexpression or knockdown approaches and this should be done as many in the field find that ChREBPbeta turns over so rapidly that it is impossible to see in western blots.

Although the half-life of CHREBP β is about 30 minutes¹⁰, a number of antibodies can detect CHREBP β protein. In addition to the antibody (58069S, CST) we used for this study, other antibodies were also found to be able to detect CHREBP β , for examples: NBP2-92977 from Novus Biologicals which has been used in the study¹¹. LS-C409190 from LS-BIO(<https://www.lsbio.com/antibodies/mlxipl-antibody-chrebp-antibody-ip-wb-western-ls-c409190/421555>) can also detect CHREBP β .

In the revised manuscript, we have used shRNA to knockdown the expression of CHREBP β (Figure 8d). The antibody used in this study clearly was able to show a decrease in CHREBP β proteins in shRNA-transfected cells in a dose-dependent manner to puromycin selection, which indirectly validated the antibody.

In addition, there are a few claims that seem overstated: 1) On page 5, the authors claim that “Loss of Runx1 fully reversed the phenotypes observed in K43M tissues, indicating that RUNX1 is the major downstream effector of CDK6 in DNL.” This is not necessarily the case - Runx1 mutation is a suppressor of Cdk6 loss of function, which could be due to many possible indirect effects. 2) “RUNX1 is a major downstream effector of K43M in DNL.” This should be re-written, as K43M is not a natural effect, therefore nothing can be downstream of it.

The sentence has been re-written as “RUNX1 is a major downstream effector of CDK6 in negative regulation of DNL” (now on page 7, line 179-180).

In addition, it is not clear to the readers why the authors use Runx1 or why this would reverse the phenotype of Cdk6 loss of function – this should be explained in the introduction or as an introduction to the results presented in the results section.

In the revised manuscript, we have inserted sentence and made it clear as “among the known targets of CDK6, RUNX1, a transcriptional factor” on page 6, line 150.

The numbering for mouse ChREBP is off by 2 base pairs – the numbering in the manuscript is for rat. Also, the sequence in human is slightly different in this region – this should be discussed. The phosphor-acceptor site for ChREBPbeta, S568, is also the acceptor site for AMP kinase, which has the same phenotype as the putative Cdk6 phosphorylation. Since the authors also see an increase in AMP kinase activity, the authors should design experiments to distinguish between the possibilities of Cdk6 phosphorylation and AMP kinase phosphorylation with *in vitro* assays.

Following the reviewer’s suggestion, we have performed proteomic analysis. The Mass Spectrometer data that support the findings of this study are available in MassIVE with the identifier (doi:10.25345/C57D2QJ4C) (reference number: MassIVE MSV000093063. The new figures are presented in Figure 7 and supplementary Figures 9 and 10. The description about the analyses and results are illustrated on page 11, lines 269-306. The significance of the phosphorylation sites is indicated on page 15-16, line 391-421.

S568 is not phosphorylated by CDK6 *in vitro* assay.

In addition, there are a number of minor issues that should be addressed:

1) The molecular weights of the proteins should be marked in all the western blots

Molecular weights are marked in all Western blots.

2) Are there any Runx1 binding sites in the ChREBP promoters?

Yes.

Employed eukaryotic promoter database (<https://epd.epfl.ch//index.php>) and ALGGEN PROMO program (http://algggen.lsi.upc.es/cgi-bin/promo_v3/promo/promoinit.cgi?dirDB=TF_8.3), a virtual laboratory for the study of transcription factor binding sites in DNA sequences, we found that mouse *Mlxip1* promoter region contains two RUNX1 consensus sequences (TGTGGT) within 1.5kb of the transcriptional start site, suggesting RUNX1 could bind to the *Mlxip1* promoter region and directly regulate its transcriptional level.

3) The first paragraph of the introduction needs references.

The references are cited on page 2, lines 50-51.

References

1. Hou, X., *et al.* CDK6 inhibits white to beige fat transition by suppressing RUNX1. *Nat Commun* **9**, 1023 (2018).
2. Tang, Y., *et al.* Adipose tissue mTORC2 regulates ChREBP-driven de novo lipogenesis and hepatic glucose metabolism. *Nat Commun* **7**, 11365 (2016).
3. Liu, J., *et al.* Monounsaturated fatty acids generated via stearoyl CoA desaturase-1 are endogenous inhibitors of fatty acid amide hydrolase. *Proc Natl Acad Sci U S A* **110**, 18832-18837 (2013).
4. Iqbal, N.J., *et al.* Cyclin-dependent kinase 4 is a preclinical target for diet-induced obesity. *JCI Insight* **3**(2018).
5. Lagarrigue, S., *et al.* CDK4 is an essential insulin effector in adipocytes. *J Clin Invest* **126**, 335-348 (2016).
6. Lagarrigue, S., *et al.* CDK4 is an essential insulin effector in adipocytes. *J Clin Invest* **133**(2023).
7. Cao, H., *et al.* Identification of a lipokine, a lipid hormone linking adipose tissue to systemic metabolism. *Cell* **134**, 933-944 (2008).
8. Huo, Y., *et al.* Targeted overexpression of inducible 6-phosphofructo-2-kinase in adipose tissue increases fat deposition but protects against diet-induced insulin resistance and inflammatory responses. *J Biol Chem* **287**, 21492-21500 (2012).
9. Hu, M.G., *et al.* CDK6 kinase activity is required for thymocyte development. *Blood* **117**, 6120-6131 (2011).
10. Uyeda, K. & Repa, J.J. Carbohydrate response element binding protein, ChREBP, a transcription factor coupling hepatic glucose utilization and lipid synthesis. *Cell Metab* **4**, 107-110 (2006).
11. Dentin, R., *et al.* Glucose 6-phosphate, rather than xylulose 5-phosphate, is required for the activation of ChREBP in response to glucose in the liver. *J Hepatol* **56**, 199-209 (2012).

REVIEWER COMMENTS

Reviewer #1 (Remarks to the Author):

In the revision version of the manuscript, the authors have addressed some of the concerns of this reviewer accordingly with the critics and suggestions. There are, however, some concerns that still need to be addressed.

1. In response to the point 2 of the previous review, it was asked to perform de novo lipogenesis experiments in the liver, however, the experiment that it is shown in sup. figure 4 cannot be significant because the low number of mice that was used (3 mice). Moreover, the labeled TG of adipose tissue should also be quantified in the same mice.
2. Concerning the experiments on DNL in response to a high fat diet, this reviewer is not convinced about the response of the authors to the question (point 4). What are the substrates that are used to increase DNL under HFD? They argue that they are independent of glucose. They show that some enzymes of DNL are increased in the K35M mice, but this reviewer does not understand what are the physiological consequences.
3. The results of the western blot analysis, as shown in the figure 3B, are not properly quantified. The expression of Glut-1, ACLY, ACC1, and FASN is indeed decreased in the K43M, and increased back in KR. This suggests an opposite role of CDK6 in the liver, relative to adipose tissue. Moreover, these results suggest that CDK6 also have a function in lipogenesis in the liver, which is also dependent on RunX1.
4. Contrary to the description in the text, there is no data on CDK6 expression in normal mice in response to HFD in sup. fig. 7a.
5. In general, the quality of the western blots could be much improved. Some examples are fig 8g, 8a, 6f (WB p-ACC1 or WB CHREBPb), 6g, 5a (mTOR), 2h (SCD1).
6. The authors have not addressed the question of why mTOR regulation is opposite in WAT and liver in K43M. This is an important observation that could be relevant for the conclusions of the study.

Reviewer #2 (Remarks to the Author):

The manuscript is much improved, and this reviewer appreciates the effort to address my concerns – most of which were met. There are a few minor things that could improve the final manuscript.

At the start of the paper it is still not clear why the authors are knocking out Runx1. The generation of Runx^{-/-} still has no context in the first paragraph of the results – it comes totally out of the blue for the reader who has not read the authors' previous papers. One or two sentences to explain why you are starting here is really necessary.

On page 11, Line 285 - "Multiple CDK6-specific phosphorylation sites were detected in AMPK-a1, and AMPK-a2 (Supplementary Fig. 10b). Among them, S47, S59, S65, and S71 are located within its N-terminus" I think this refers to ChREBP

In terms of possible compensatory mechanisms for the knockdown of ChREBP and the SCD1 result, the authors might consider MondoA and Srebp-1c.

Point-by-point response (NCOMMS-22-44553A)

We thank the reviewers for their further comments and constructive suggestions. We have addressed their concerns in the revised manuscript. The following is the point-by-point response.

Reviewer #1

In the revision version of the manuscript, the authors have addressed some of the concerns of this reviewer accordingly with the critics and suggestions. There are, however, some concerns that still need to be addressed.

1. In response to the point 2 of the previous review, it was asked to perform de novo lipogenesis experiments in the liver, however, the experiment that it is shown in sup. figure 4 cannot be significant because the low number of mice that was used (3 mice). Moreover, the labeled TG of adipose tissue should also be quantified in the same mice.

Response: In response to the point 2 of the reviewer on the previous version of the manuscript, we performed additional *in vivo* de novo lipogenesis (DNL) in the liver of 6 *WT* and 7 *K43M* conscious mice. The *WT* group had 3 male and 3 female, and the *K43M* group had 3 male and 4 female. We did not include more mice in both groups because these experiments were to confirm the results that DNL in the liver was not significantly different between *WT* and *K43M* mice, a conclusion we made in the original submission based on the findings already presented in Figures 2h, 2j, 3b, and 3d. We fully understand the concerns of the reviewer that the number of mice was low if the results of male and female are plotted separately. Therefore, we combined the male and female mice to bring the number of *WT* and *K43M* mice to 6 and 7, respectively, and reanalyzed the data. The results show that there is no statistical difference in DNL in the liver between *WT* and *K43M* mice (Supplementary Figure 3f), confirming the findings presented in Figure 2 and 3. We have also revised **Figure 1a-c and Supplementary Figure 3** to accommodate the additional results obtained from increased number of male and female mice in both *WT* and *K43M* groups.

Supplementary Figure 3 and Supplementary Figure 4 in the previous submission has been combined into new Supplementary Figure 3. The numbering of other supplementary Figures has been changed accordingly.

2. Concerning the experiments on DNL in response to a high fat diet, this reviewer is not convinced about the response of the authors to the question (point 4). What are the substrates that are used to increase DNL under HFD? They argue that they are independent of glucose. They show that some enzymes of DNL are increased in the *K35M* mice, but this reviewer does not understand what the physiological consequences are.

Response: We agree with the reviewer that feeding mice with high sucrose diet will better reflect de novo lipogenesis in response to high carbohydrate diet. Our focus in this study is high fat diet-induced de novo lipogenesis. The substrates under HFD could include carbohydrates (glucose

and fructose), proteins (amino acids), acetate, glycerol, and alcohol so we cannot conclude that it is glucose independent. We are thankful for the reviewer to point this out.

We have discussed the possible physiological consequence of increased DNL on page 2-3, line 57-69, which reads: “Glucose is taken up and converted to citrate through glycolysis and TCA cycle. By sequential action of ACLY, ACC, and FASN, citrate is converted to acetyl-CoA, malonyl-CoA, and finally palmitate. Stearoyl-CoA desaturase-1 (SCD) then converts palmitate to palmitoleate that mediates the insulin-sensitizing effects of DNL in WAT^{1,2}. In addition, DNL in WAT produces other bioactive fatty acids such as fatty acid ester of hydroxyl fatty acids (FAHFAs), a new class of lipids that are important contributors to the improvement of IR¹⁻³ and suppression of inflammation in adipose tissues (ATs) by stimulating intestinal cells to secrete glucagon-like peptide 1 (GLP-1), pancreas to release insulin, immune cells to reduce inflammatory cytokine production, as well as enhancing glucose transport in the cells⁴⁻⁷. Paradoxically, DNL in WAT is downregulated in obesity^{1,3,8} and it is known that restoring DNL in WAT selectively reverts obesity-induced IR^{1,8}. In contrast, DNL in the liver appears to be increased in obesity, and is believed to promote IR, lipotoxicity, and hepatic steatosis⁹. Together, this evidence suggests that reduction of DNL in WAT but its increase in the liver is contributing factors to systemic IR and other metabolic diseases”.

3. The results of the western blot analysis, as shown in the figure 3B, are not properly quantified. The expression of Glut-1, ACLY, ACC1, and FASN is indeed decreased in the K43M, and increased back in KR. This suggests an opposite role of CDK6 in the liver, relative to adipose tissue. Moreover, these results suggest that CDK6 also have a function in lipogenesis in the liver, which is also dependent on RunX1.

Response: We have re-quantified the expression of these proteins, and confirmed the results that the recovery of these proteins in KR livers did not reach a statistically significant level, which is consistent with the fact that *Adipoq-Cre* was used in our study that only delete *Runx1* in mature adipocytes but not in the liver. We have revised these findings in the revised manuscript. It now reads: “In contrast, these DNL-related proteins were not increased, but decreased in the liver of *K43M* mice, even though only ACLY reached statistical significance (**Fig. 3b, d**). DNL-related proteins were slightly but not statistically significantly recovered in KR livers, consistent with the fact that *Adipoq-Cre* only delete *Runx1* in mature adipocytes but not in the liver. Thus, the role of Runx1 in CDK6-mediated DNL in the liver, if any, is unclear currently” (page 7, lines 180-184).

4. Contrary to the description in the text, there is no data on CDK6 expression in normal mice in response to HFD in sup. fig. 7a.

Response: The expression of CDK6 under NCD and HFD has been published in our previous paper, which was cited (reference # 22) in the text on page 7, line 173. We have also included Sup. Fig. 7a to display the expression of CDK6 in the liver.

5. In general, the quality of the western blots could be much improved. Some examples are fig 8g, 8a, 6f (WB p-ACC1 or WB CHREBP β), 6g, 5a (mTOR), 2h (SCD1).

Response: The quality of the western blots has been improved in the Figures 8g, 8a, 6f (WB p-ACC1 or WB CHREBP β), 6g, 5a (mTOR), 2h (SCD1) by adjusting exposure time. We have also replaced Figure 6h with blots of higher quality from a different experiment.

6. The authors have not addressed the question of why mTOR regulation is opposite in WAT and liver in K43M. This is an important observation that could be relevant for the conclusions of the study.

Response: In the revised manuscript, we have discussed that “The exact mechanism for the differential effect of the different signaling pathways involved may account for the difference of CDK6 effects on DNL between VAT and the liver is unknown. One possibility could be the different signaling pathways involved in the two organs. For instance, phosphorylation levels of mTOR and AMPK α was reduced in adipose tissues, while those levels were increased in the livers (Figure 5). This dichotomous regulation of mTOR and AMPK α phosphorylation could be an underlying reason for the opposite regulation of DNL in the VAT and the liver.” (page 16, lines 397-401).

Reviewer #2 (Remarks to the Author):

The manuscript is much improved, and this reviewer appreciates the effort to address my concerns – most of which were met. There are a few minor things that could improve the final manuscript.

At the start of the paper it is still not clear why the authors are knocking out Runx1. The generation of Runx $^{-/-}$ still has no context in the first paragraph of the results – it comes totally out of the blue for the reader who has not read the authors’ previous papers. One or two sentences to explain why you are starting here is really necessary.

Response: We have added a rationale for targeting Runx1 in the revised manuscript on page 4, lines 98-104. It reads: “Our previous studies have demonstrated that CDK6 has a positive role in induction of obesity by suppressing runt related transcription factor 1 (*Runx1*)^{10,11} that is known to be frequently mutated in human leukemia and play a role in hematopoiesis¹². To genetically ablate *Runx1* in VAT of *WT/K43M* mice (**Supplementary Fig. 1**), we crossed *WT/K43M* mice with *Runx1* mutant mice (*Runx1*^{fl/fl}) that bear two *loxP* sites flanking *exon 4* of the *Runx1* allele¹³ (**Supplementary Fig. 2b**). The resultant *WT;Runx1*^{fl/fl} / *K43M;Runx1*^{fl/fl} mice were then crossed with *Adipoq-Cre* mice¹⁴ to delete *Runx1* in mature adipocytes¹⁰ (**Supplementary Fig. 1**).”

On page 11, Line 285 - “Multiple CDK6-specific phosphorylation sites were detected in AMPK-a1, and AMPK-a2 (Supplementary Fig. 10b). Among them, S47, S59, S65, and S71 are located within its N-terminus” I think this refers to ChREBP.

Response: The mistake has been corrected.

In terms of possible compensatory mechanisms for the knockdown of ChREBP and the SCD1 result, the authors might consider MondoA and Srebp-1c.

Response: The possible compensatory mechanisms by MondoA and Srebp-1c in the absence of CHREBP have been discussed on page 15, line 381, and page 18, line 443.

1. Cao, H., *et al.* Identification of a lipokine, a lipid hormone linking adipose tissue to systemic metabolism. *Cell* **134**, 933-944 (2008).
2. Yang, Z.H., Miyahara, H. & Hatanaka, A. Chronic administration of palmitoleic acid reduces insulin resistance and hepatic lipid accumulation in KK-Ay Mice with genetic type 2 diabetes. *Lipids Health Dis* **10**, 120 (2011).
3. Eissing, L., *et al.* De novo lipogenesis in human fat and liver is linked to ChREBP-beta and metabolic health. *Nat Commun* **4**, 1528 (2013).
4. Vijayakumar, A., *et al.* Absence of Carbohydrate Response Element Binding Protein in Adipocytes Causes Systemic Insulin Resistance and Impairs Glucose Transport. *Cell Rep* **21**, 1021-1035 (2017).
5. Yore, M.M., *et al.* Discovery of a class of endogenous mammalian lipids with anti-diabetic and anti-inflammatory effects. *Cell* **159**, 318-332 (2014).
6. Lodhi, I.J., Wei, X. & Semenkovich, C.F. Lipoexpediency: de novo lipogenesis as a metabolic signal transmitter. *Trends Endocrinol Metab* **22**, 1-8 (2011).
7. Smith, U. & Kahn, B.B. Adipose tissue regulates insulin sensitivity: role of adipogenesis, de novo lipogenesis and novel lipids. *J Intern Med* **280**, 465-475 (2016).
8. Huo, Y., *et al.* Targeted overexpression of inducible 6-phosphofructo-2-kinase in adipose tissue increases fat deposition but protects against diet-induced insulin resistance and inflammatory responses. *J Biol Chem* **287**, 21492-21500 (2012).
9. Postic, C. & Girard, J. Contribution of de novo fatty acid synthesis to hepatic steatosis and insulin resistance: lessons from genetically engineered mice. *J Clin Invest* **118**, 829-838 (2008).
10. Hou, X., *et al.* CDK6 inhibits white to beige fat transition by suppressing RUNX1. *Nat Commun* **9**, 1023 (2018).
11. Biggs, J.R., Peterson, L.F., Zhang, Y., Kraft, A.S. & Zhang, D.E. AML1/RUNX1 phosphorylation by cyclin-dependent kinases regulates the degradation of AML1/RUNX1 by the anaphase-promoting complex. *Mol Cell Biol* **26**, 7420-7429 (2006).
12. Orkin, S.H. Development of the hematopoietic system. *Curr Opin Genet Dev* **6**, 597-602 (1996).
13. Taniuchi, I., *et al.* Differential requirements for Runx proteins in CD4 repression and epigenetic silencing during T lymphocyte development. *Cell* **111**, 621-633 (2002).
14. Cristancho, A.G. & Lazar, M.A. Forming functional fat: a growing understanding of adipocyte differentiation. *Nat Rev Mol Cell Biol* **12**, 722-734 (2011).

REVIEWERS' COMMENTS

Reviewer #1 (Remarks to the Author):

The authors have properly addressed the concerns of this reviewer. I do not have any further concern or critic. The manuscript is now of very good quality.

Reviewer #2 (Remarks to the Author):

all my concerns were addressed